# Plasticity in inhibitory networks improves pattern separation in early olfactory processing

Shruti Joshi [1,2] ✉, Seth Haney[2], Zhenyu Wang[3], Fernando Locatelli[4], Hong Lei[5], Yu Cao[6], Brian Smith[5] & Maxim Bazhenov [2] ✉

Distinguishing between nectar and non-nectar odors is challenging for animals due to shared compounds and varying ratios in complex mixtures. Changes in nectar production throughout the day and over the animal's lifetime add to the complexity. The honeybee olfactory system, containing fewer than 1000 principal neurons in the early olfactory relay, the antennal lobe (AL), must learn to associate diverse volatile blends with rewards. Previous studies identified plasticity in the AL circuits, but its role in odor learning remains poorly understood. Using a biophysical computational model, tuned by in vivo electrophysiological data, and live imaging of the honeybee's AL, we explored the neural mechanisms of plasticity in the AL. Our findings revealed that when trained with a set of rewarded and unrewarded odors, the AL inhibitory network suppresses responses to shared chemical compounds while enhancing responses to distinct compounds. This results in improved pattern separation and a more concise neural code. Our calcium imaging data support these predictions. Analysis of a graph convolutional neural network performing an odor categorization task revealed a similar mechanism for contrast enhancement. Our study provides insights into how inhibitory plasticity in the early olfactory network reshapes the coding for efficient learning of complex odors.

A honeybee can locate nectar-producing flowers by detecting floral aromas composed of many volatile compounds[1,2]. However, nectar-producing and non-producing flowers contain many of the same compounds, making it difficult for the honeybee to determine the map between odor sensing and reward prediction[3]. This task is further complicated by the fact that nectar production may change within days and across locations, potentially many times within the foraging lifetime of a honey bee. Although the olfactory coding space is large (close to 1000 projection neurons in the honeybee antennal lobe[4,5]), mapping the sensory environment requires learning and relearning the association of reward with variable blends of volatile compounds. Nevertheless, honeybees can adapt to the variances in their environment owing to their keen ability to discriminate between a wide range of olfactory stimuli[6–9].

Different forms of plasticity are a ubiquitous feature of the early olfactory processing in the brains of both mammals and insects. In the neural networks of the mammalian olfactory bulb (OB) and insect antennal lobe (AL), both nonassociative (unsupervised) and associative (supervised) forms of plasticity have been described[8,10–12]. Together with the now well-established similarities in anatomical connectivity within the AL and OB networks[13,14], it is clear that early olfactory processing in these phylogenetically very different groups of animals works in much the same way to augment olfactory processing.

Studies across different insect species have revealed conserved mechanisms of plasticity in early olfactory processing. In Drosophila, prolonged odor exposure leads to selective potentiation of inhibitory synapses between local interneurons (LNs) and projection neurons (PNs), resulting in habituation[15]. In moths, associative conditioning modifies both excitatory and inhibitory connections within the AL network, enhancing responses to behaviorally relevant odors[16]. Work in locusts has demonstrated that plasticity in the antennal lobe can shape representations to improve stimulus discrimination[17,18]. In honeybees, both associative and non-associative forms of plasticity have been shown to modify AL network dynamics, with

[1]Department of Electrical and Computer Engineering, University of California San Diego, La Jolla, CA, USA. [2]Department of Medicine, University of California San Diego, La Jolla, CA, USA. [3]Department of Electrical, Computer and Energy Engineering, Arizona State University, Tempe, AZ, USA. [4]Facultad de Ciencias Exactas y Naturales, Universidad de Buenos Aires, Instituto de Fisiología, Biología Molecular y Neurociencias, CONICET, Buenos Aires, Argentina. [5]School of Life Science, Arizona State University, Tempe, AZ, USA. [6]Department of Electrical and Computer Engineering, University of Minnesota, Minneapolis, MN, USA. ✉e-mail: s4joshi@ucsd.edu; mbazhenov@health.ucsd.edu

computational models suggesting that such modifications can enhance discrimination between similar odors[9,12,19–21]. Together, these studies highlight inhibitory synaptic plasticity as a conserved mechanism for optimizing early olfactory processing across insect species.

In this new study, we combined Ca$^{2+}$ imaging from the honeybee AL with biophysical computational modeling to characterize the role of inhibitory synaptic plasticity in the AL in improving odor discrimination after olfactory learning. We found that representation of chemical compounds shared between rewarded and habituated odors are suppressed while distinct ones are enhanced after learning to increase contrast and reduce overlap between odor representations. This change in representation requires both associative and nonassociative plasticity on the inhibitory synapses between AL neurons. Analysis of the Ca$^{2+}$ imaging data revealed a change in odor representations after learning, supporting the model's prediction. Applying these ideas to the Artificial Neural Network (ANN) models trained to perform the odor categorization task revealed principles of contrast enhancement applicable to machine learning. In sum, we demonstrate the role of inhibitory synaptic plasticity in the AL for effective odor discrimination and propose a novel computationally efficient mechanism for performing categorization tasks.

## Results

### Differential conditioning created distinct representations of odors in vivo

In our previous work[12] we used Ca$^{2+}$ imaging to test how the representation of synthetic odor blends changes in the honeybee AL after differential conditioning (Fig. 1A). Two synthetic classes of odors were based on varieties of the common snapdragon flower (*A. majus*): Potomac Pink (PP) and Pale Hybrid (PH)[22]. Each class was a blend of volatile chemicals that recapitulated mean and variance of naturally occurring PH and PP flowers. These two classes contained the same volatile chemicals at different ratios that makes distinguishing them a difficult task.

We found that odors within each class created distinct representations in the AL via activation of different sets of glomeruli (Fig. 1B, C). Further, these representations became more distinct after differential training. Specifically, in absolute conditioning, honeybees were rewarded after responding to PH odors; in differential conditioning honeybees were also habituated to the PP odors[12] (Fig. 1E). We found that the correlation between the representations of PH and PP odors decreased after differential conditioning (Fig. 1D).

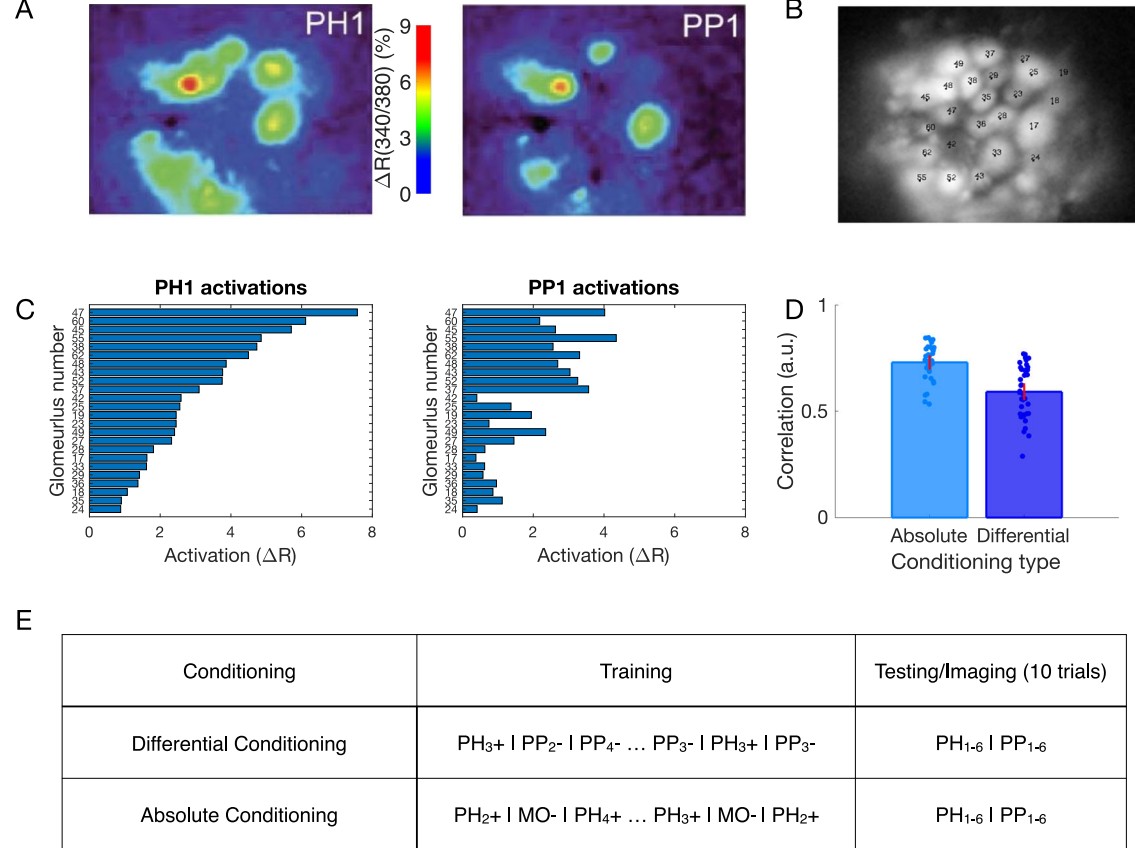

**Fig. 1 | Olfactory representations in the honeybee AL. A** Calcium imaging responses elicited by two different odors (PH1 and PP1) in a representative bee. Here, PH1 and PP1 belong to different classes of odors, with PH odors being presented along with a reward (rewarded odors) and PP odors presented without any reward (habituated odors). **B** Position of the glomeruli within the antennal lobe, for which calcium imaging responses were measured. **C** Mean activation across the different glomeruli elicited by the PH1 and PP1 odors. Glomeruli are ordered such that the glomeruli most activated by PH1 at the top and the glomeruli least activated by PH1 at the bottom. The same ordering is used for PP1 odor as well to highlight the difference in the activation patterns between the two odors. **D** Correlation of glomeruli activation between different odor classes (PH and PP) decreases after differential conditioning compared to absolute conditioning (Wilcoxon signed rank

test, $p = 4.8e-8$, $n = 7$ bees with absolute conditioning and 10 bees with differential conditioning). This indicates that the glomerular representation gets more separable after differential conditioning. The error bars here show standard error of mean. Individual data points show correlation between each pair of PH and PP odors (PH1 vs PP1, PH1 vs PP2 etc). **E** Honeybee training protocol for differential and absolute conditioning. In differential conditioning, the bee was trained with rewarded (PH) and habituated (PP) odors presented in a randomized order. For absolute conditioning, the bee was trained with only rewarded odors. Mineral Oil (MO) was presented to the bee without any reward during absolute conditioning. Each trial lasts for 4 s, with 500 ms of odor presentation (Data from ref. 12). The same training protocol was used for training the computational model.

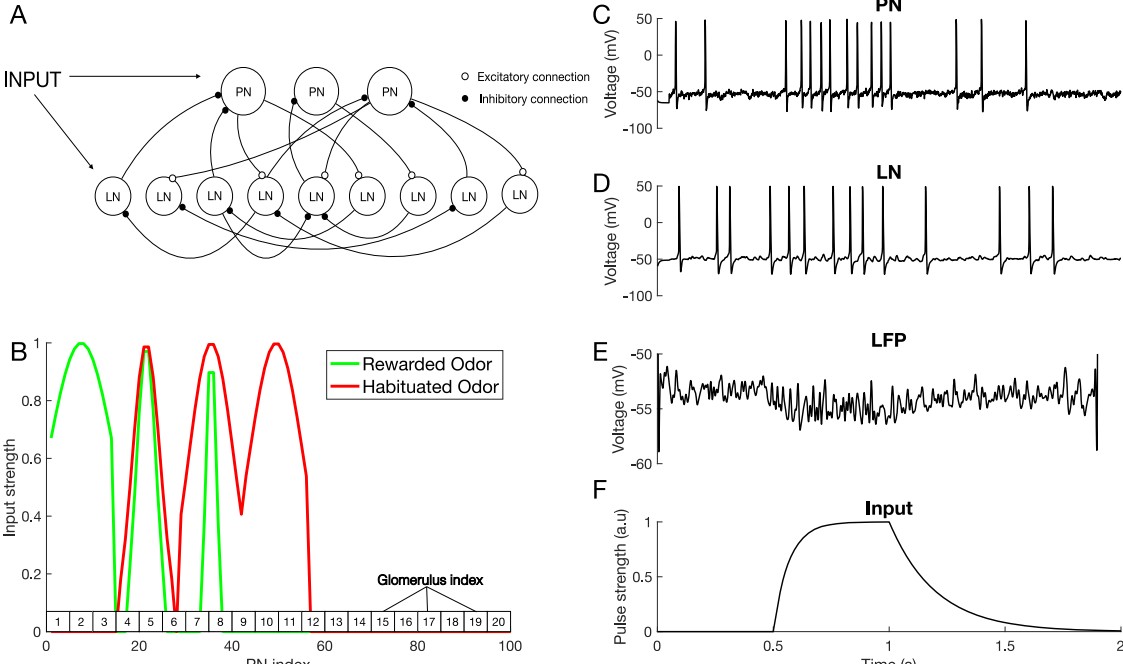

**Fig. 2 | Computational model of the antennal lobe. A** Architecture of the AL network model, including excitatory PNs and inhibitory LNs (see "Methods"). **B** Spatial pattern of odor stimuli for rewarded and habituated odors. The PNs are split into seven groups, each representing one odor percept: Percept 1 = [PN1–PN14]; Percept 2 = [PN15–PN28]; …. Each simulated odor activates three selected percepts. In this experiment, the rewarded odor (green) activates percepts 1, 2, and 3, while the habituated odor (red) activates percepts 2, 3, and 4. The green and red traces display the activation levels of individual neurons within each percept, which are defined by Gaussian curves of varying widths. Note that a given odor presentation may activate many or only a few neurons in a percept. The rectangles on top of the *X*-axis show the glomerulus indexes, which are used to set up anatomical connectivity between LNs and PNs (see "Methods"). Responses of a representative PN (**C**) and LN (**D**) to an odor input in the untrained network. **E** Averaged PN activity (Local Field Potential). **F** The 500 ms time modulated current pulse which was provided as an inputs to individual neurons during odor stimulation (Gaussian noise was added on top of this input—see "Methods").

## Computational model of the mechanisms of differential conditioning

Here, we sought to investigate the mechanisms that produce differential learning between complex odors in the honeybee AL using a computational model. The model was a network of Hodgkin-Huxley type excitatory principal neurons (PNs) and inhibitory local neurons (LNs), representing honeybee AL network, and it was adapted from our previous work[21] (see "Methods", Fig. 2A). This model was previously tuned to capture characteristic features of the AL dynamics, e.g., odor triggered LFP oscillations[23], and thus it enables us to study mechanisms of plasticity without compromising realistic AL dynamics which may play a role in shaping representations.

To model reward learning—appetitive conditioning—we simulated activity-dependent presynaptic facilitation at both the LN-PN and LN-LN synapses (see "Methods") inspired by the finding that octopamine receptors activated by reward signal are localized on the inhibitory LNs in the AL[24]. Octopamine is needed for appetitive olfactory learning[7,25] and it has been shown to regulate inhibitory connections within the AL, affecting odor representations[26]. Here we focus on AmOA1 octopamine receptor[27] expressed alongside GABA receptors in the honeybee AL[24]. In honeybees[28] and fruit flies[29], octopamine binding to this receptor leads to an increase in intracellular Ca[2+] that primes adenylyl cyclase to enhance cAMP and cAMP-dependent protein kinase activation that further regulates synaptic plasticity[30–33]. Therefore, we simulated the appetitive conditioning by increasing the synaptic weights of LN-PN and LN-LN synapses as the presynaptic LN was activated by the conditioned odor. Habituation was modeled as post-synaptic facilitation of LN-PN and LN-LN synapses, i.e., it depended on activity of postsynaptic LNs or PNs, based on the finding that habituation causes LN-PN facilitation in the fruit fly AL[34,35].

The spiking activity of a representative PN and LN are shown in Fig. 2C, D. Figure 2E shows the local field potential (LFP) obtained by averaging membrane voltages across the PN population and Fig. 2F shows temporal structure of the input current pulse to the neurons during odor stimulation. The amplitude of this current pulse to each neuron depends on the odor currently being presented.

To account for the fact that natural odors are composed of a blend of distinct volatile chemicals, we created inputs to the model combining activation of multiple distinct odor "percepts". A percept can be thought of as AL activation induced by specific chemical component of the odor (see ref. 12 and Fig. 5A below). Each percept in the model was represented by a distinct population of "virtual" ORNs making connections to simulated LNs and PNs. This effectively divided all the LNs and PNs into several percept groups (Fig. 2B). A percept in the model could include more than one glomerulus consistent with an observation that a chemical can activate more than one glomerulus, each at a different intensity, in vivo[4]. As with real floral odors[12], model odors were composed of multiple overlapping percepts creating a difficult discrimination task.

While our AL model dynamics reasonably match in vivo data (see Supplementary Fig. 1), PN firing rates in the model are slightly lower than those observed in vivo, particularly during odor stimulation and post-odor response. It is important to note, however, that in vivo recordings are naturally biased towards more active neurons. Additionally, model PN responses exhibit somewhat simplified temporal patterns, which is related to the lack of access to the simultaneous ORN recordings. Therefore, we used simplified approximations of the odor input, similar to our previous work in locusts[36,37], moths[38], and honeybees[21].

## Differential training places stimuli further apart in the coding space

We trained our model using differential conditioning, where one class of odors $A = (A1, A2, …)$ was rewarded and another $B = (B1, B2, …)$ was habituated. Within each class, different odors were simulated by adjusting

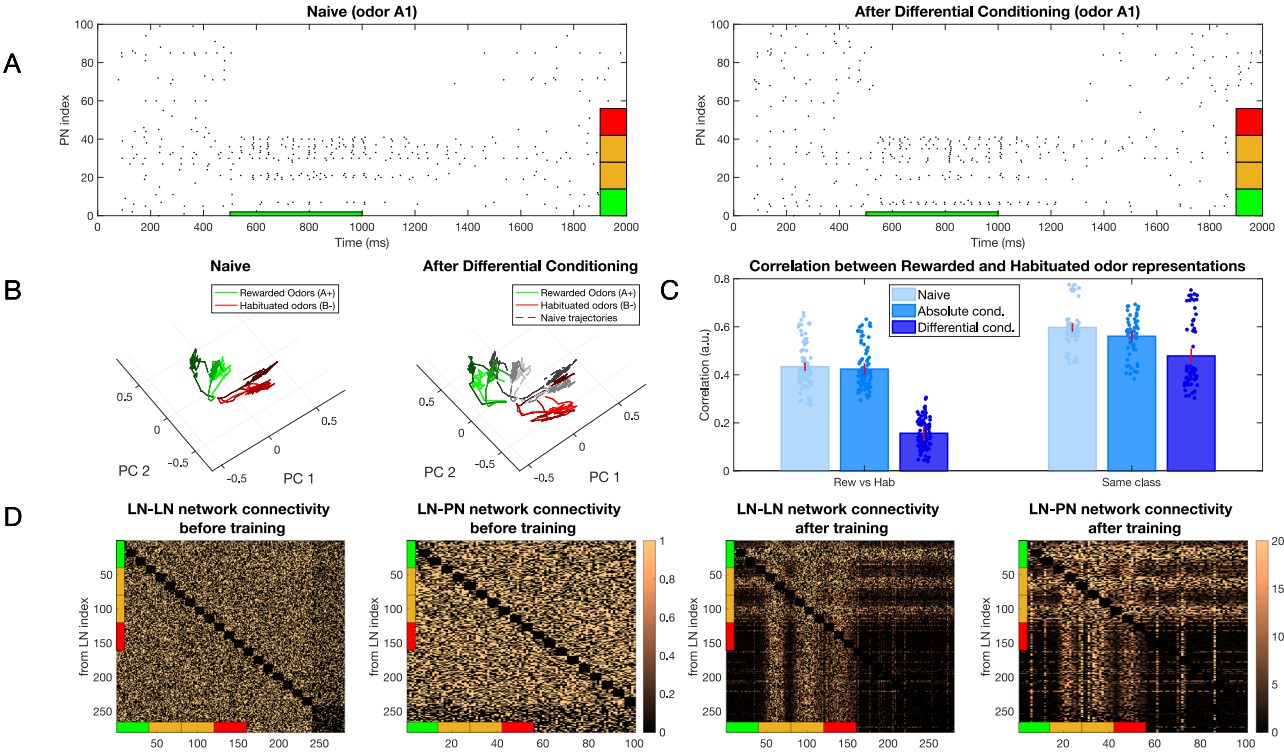

**Fig. 3 | Change in representation of simple odors after learning. A** Raster plot of the PN population response to rewarded odor (odor *A1*) before (left) and after (right) differential conditioning. Note, that activity of PNs activated by both rewarded and habituated odors (denoted by orange percepts) decreased while the activity of PNs activated uniquely by the rewarded odor (denoted by the green percept) increased. Red indicates percepts uniquely activated by the habituated odor. **B** Dynamical trajectory of the spatiotemporal odor responses in two-dimensional (2D) PCA space. The green lines show trajectories for 3 Rewarded Odors and the red ones for 3 Habituated Odors. After training, the trajectories of the rewarded and habituated odors shift away from each other. Black lines on the right plot indicate naive trajectories. **C** Left panel shows the correlation between representations of rewarded and habituated odors across three conditions: naive networks (light blue), after absolute conditioning (medium blue), and after differential conditioning (dark blue). While the correlation between odor representations from different classes

remains largely unchanged after absolute conditioning compared to the naive state ($p = 0.32$), it decreases significantly following differential conditioning ($p = 6e{-}7$). The right panel illustrates correlations between representations of odors within the same class. These within-class correlations show a modest decrease after both absolute conditioning ($p = 0.1$) and differential conditioning ($p = 0.07$) relative to the naive condition. For both panels, correlations were computed pairwise between odors and averaged across all pairs, with error bars representing the standard deviation across $n = 10$ trials. Dots represent individual data points. **D** Connectivity of the inhibitory networks (LN-LN and LN-PN) before (left) and after (right) training. After differential conditioning, connectivity exhibits a grid-like structure that encodes the relationships among percepts associated with the rewarded and habituated odors. The colorbars show the final value of the weight divided by its initial value.

the activation level of individual neurons for each percept through modifications to the Gaussian width (Fig. 2B). The model network was exposed to a sequence of $N = 30$ odors total presented in a randomized order, for a total of 60 s (each odor presentation lasted 2 s). The plasticity rule was selected based on which type of odor (i.e., rewarded or habituated) was presented. We found that the training led to significant changes in the PNs' population response for the rewarded odors (Fig. 3A). Specifically, neurons associated with percepts that were unique to the rewarded odors showed enhanced firing rates after training, whereas neurons associated with percepts in the overlap between rewarded and habituated odors were strongly suppressed. For habituated odors, plastic changes led to rather minor and non-specific reduction of the PN responses. These combined effects effectively placed stimuli further apart in the AL coding space enhancing discrimination between rewarded and habituated odors.

We visualized effect of training using principal component analysis (PCA) (Fig. 3B). We found that the trajectories of the naive response showed a relatively even distribution across PCA space. In contrast, after training, the sets of trajectories representing rewarded and habituated odors were strongly separated.

Correlation analysis revealed a significant decrease in correlation after differential training when comparing rewarded vs. habituated odors and a modest decrease when comparing odors within the same class, i.e., just rewarded or just habituated odors (Fig. 3C). Note that odors from the same

class are inherently more similar to each other than odors from different classes, even in the naive condition. Both these results are in agreement with experimental data[12]. The changes to odor representation in our model were due to changes in the inhibitory network, as these are the only synapses modified in the model. Analysis of the synaptic connectivity matrix after learning revealed a characteristic structure reflecting the structure of percepts in rewarded and habituated odors (Fig. 3D).

To evaluate the robustness of our findings, we conducted a sensitivity analysis of the key model parameters. Specifically, we systematically varied the following: fast and slow GABAergic and ACh synaptic conductances between LNs and PNs, input synaptic conductances to LNs and PNs, and synaptic plasticity parameters (facilitation rate and decay time constant). We found that the model's core behavior—enhanced separation between rewarded and habituated odor representations after learning—remains qualitatively consistent across a broad range of parameter values (Supplementary Fig. 2).

Next, we individually tested the effect of associative vs. nonassociative plasticity on the ability of the trained network to distinguish between the rewarded and habituated odors by selectively applying only pre- or post-synaptic plasticity and measuring the correlation between the learned odor representations. In Fig. 4A, we present the correlation between rewarded and habituated representations under four conditions: (1) Naive, with both pre- and post-synaptic plasticity off; (2) Associative only, where

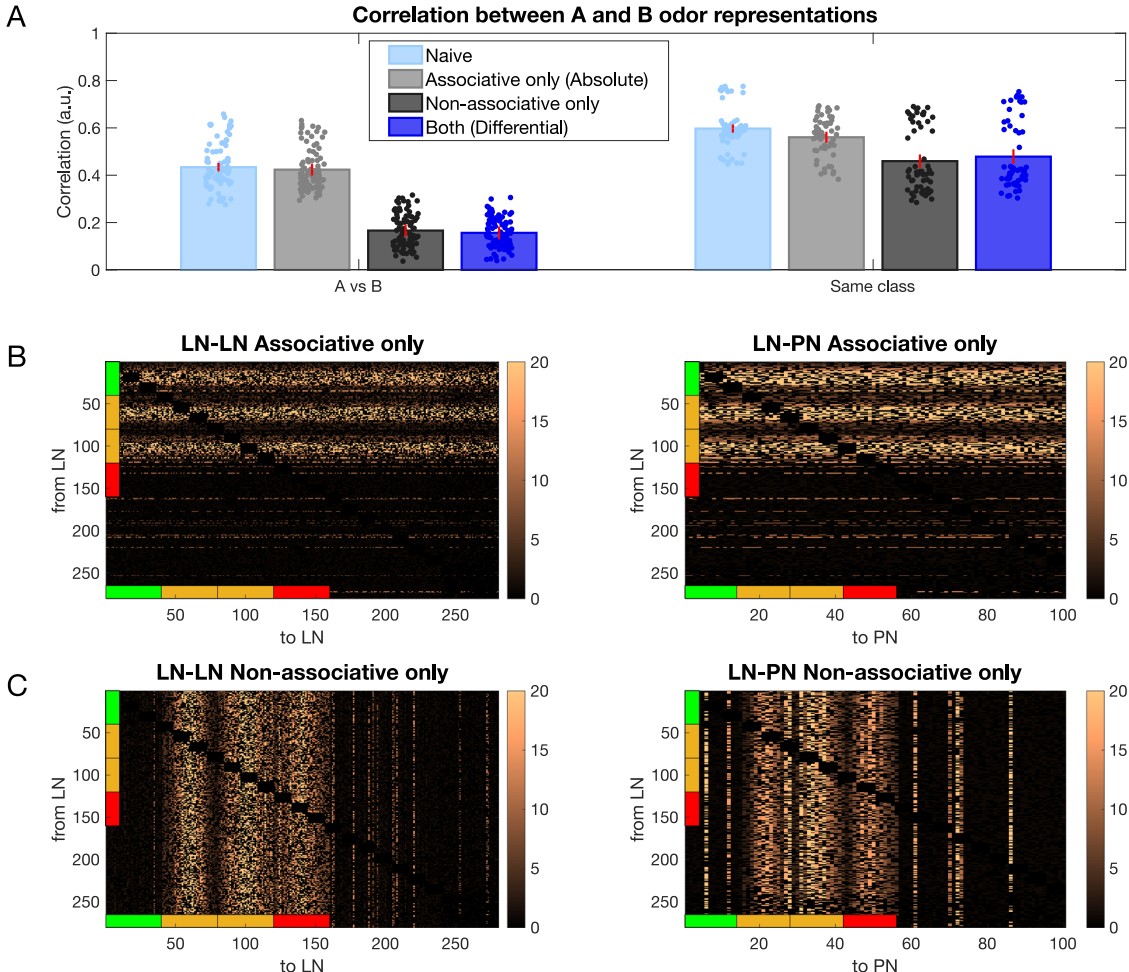

**Fig. 4 | Connectivity of the inhibitory network after associative only and non-associative only conditioning. A** Left shows the correlation between rewarded and habituated odor representations does not decrease significantly after associative (gray) conditioning ($p = 0.325$) but decreases significantly after nonassociative (black) conditioning ($p = 2e-7$) and differential (dark blue) conditioning ($p = 6e-7$) compared to naive (light blue). Right shows correlation calculated for odors belonging to the same class, with correlation not changing significantly after associative (gray) conditioning ($p = 0.1$), but decreasing modestly after non-associative (black, $p = 0.04$) and differential (dark blue) conditioning ($p = 0.07$). The error bars represent standard deviation across $n = 10$ trials. Dots represent individual data points. **B** The LN-LN and LN-PN network after associative only training. **C** The LN-LN and LN-PN network after nonassociative only training. The colorbars show the value of the weights after training divided by the initial weights.

pre-synaptic plasticity is on and post-synaptic plasticity is off (absolute conditioning); (3) Non-Associative only, with pre-synaptic plasticity off and post-synaptic plasticity on; and (4) Both, where both pre- and post-synaptic plasticity are on (differential conditioning). Compared with the naive condition, we observed a significant decrease in correlation when both associative and nonassociative plasticity were active, as well as when only nonassociative plasticity was active (Fig. 4A, left). In contrast, the associative only case did not show a significant decrease in correlation when compared with the naive condition. We also observed a modest decrease in the correlation between representations of odors from the same class across all three conditions compared with the naive case (Fig. 4A, right). Figure 4B, C illustrates the network structures of LN-LN and LN-PN connections following learning for the associative only and nonassociative only conditions, respectively. As expected, it reveals changes in only subset of connections (compare to Fig. 3D). Note that in Fig. 4B, C, bright horizontal lines (in Fig. 4B) and vertical lines (in Fig. 4C) indicate increased inhibition to certain PNs that are not directly activated by the odor. For instance, in the right panel of Fig. 4C, vertical lines between PN 60 and 80 and between PN 80 and 100 are evident. These patterns likely emerge because some PNs exhibit high background firing rates, which appear to be further enhanced after training.

The training increases the total amount of inhibition in the network. To verify that the odor-specific structure of inhibitory connections, rather than the overall increase in inhibition, aids odor discrimination, we conducted a "shuffling experiment". Specifically, we maintained an identical distribution of the weights in the shuffled networks but disrupted the structured connectivity patterns that emerged during learning. These shuffled networks, despite having an equivalent total inhibitory drive, failed to produce the between- and within-class odor decorrelation observed in the structured network (see Supplementary Fig. 3). These results support our prediction that enhanced odor discrimination arises from the precise restructuring of inhibitory circuits during learning rather than from a global increase in inhibition.

**Training with realistic odorants also produces contrast enhancement**

To ensure that the observed effects were not simply due to the specific construction of our model odors, we generated a new set of odors. In this set, the proportions of percepts were aligned with the measured proportions of volatile chemicals in each odor blend category (e.g., PH1, PH2, … and PP1, PP2, …) used in the behavioral experiments described in ref. 12. These PH and PP odor classes were generated by calculating the proportions of the chemical components present in the blends and assigning the activation of each percept accordingly (Fig. 5A, see "Methods" for details). Training these new odors using differential conditioning (rewarding PH and habituating

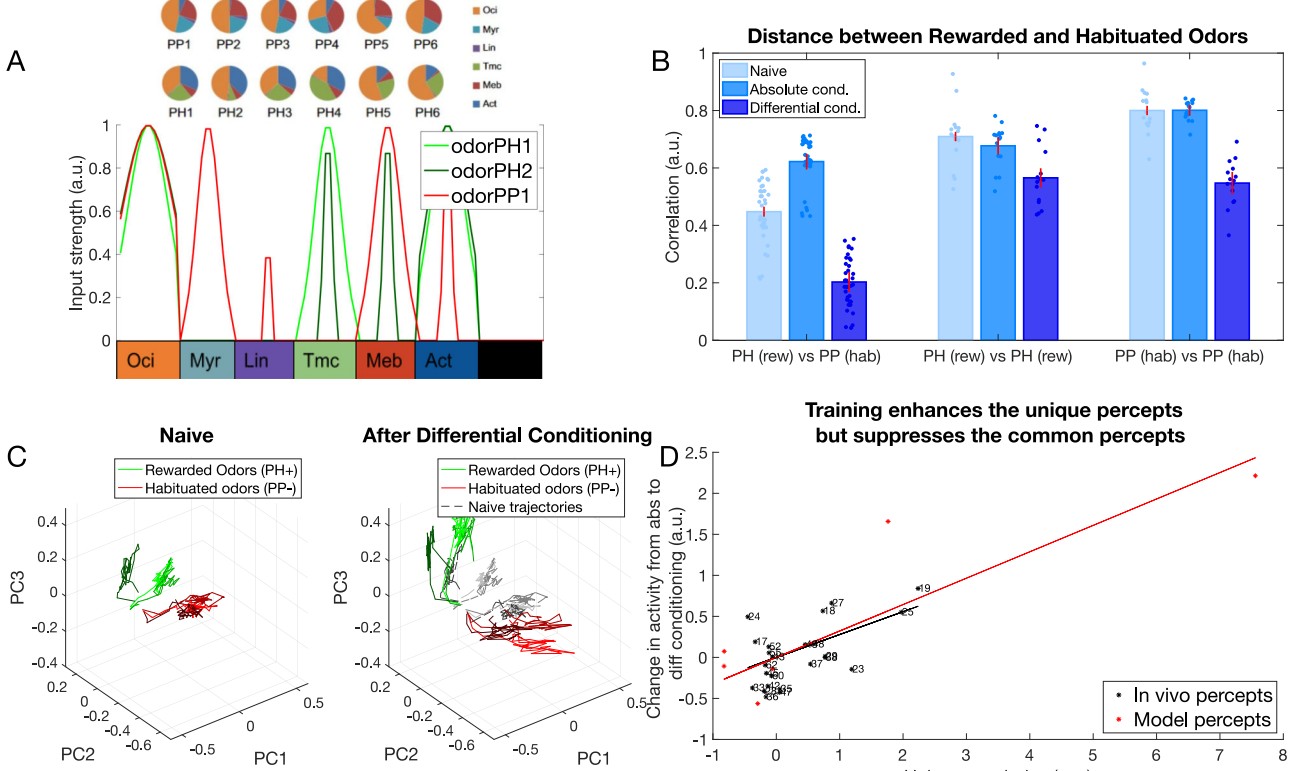

**Fig. 5 | Contrast enhancement with complex odors based on in vivo data. A** Odors obtained from in vivo *experiments* (PH,PP) converted into inputs to the model (see "Methods" for details). **B** Correlation between rewarded (PH) and habituated (PP) odor representations decreases after differential conditioning (dark blue) compared to the naive network (light blue, $p = 4.2e−29$) and absolute conditioned network (middle tone blue, $p = 8.7e−38$). Correlation between odors belonging to same rewarded class (e.g., PH1 vs. PH2, PH1 vs. PH3, …) increases slightly after absolute conditioning (middle tone blue, $p = 0.05$) and significantly after differential conditioning (dark blue, $p = 4.7e−7$) compared to the naive case. Correlation between the odors belonging to the habituated class does not decrease after absolute conditioning (middle tone blue, $p = 0.71$) but decreases significantly after differential conditioning (dark blue, $p = 1.5e−12$) compared to the naive case. These results match the experimental data from ref. 12. **C** The dynamical odor trajectory in the 3D PCA space shows the 3 rewarded (PH1, PH2, PH3) and 3 habituated odor (PP1, PP2, PP3) representations shifting in the opposite direction which indicates increased discriminability between odor representations. The error bars represent standard deviation across $n = 10$ trials. Dots represent individual data points. **D** Contrast enhancement based on $Ca^{2+}$ imaging data (black) and model (red). After differential conditioning, activity in the glomeruli/percepts which have a high uniqueness index (i.e., uniquely activated by the rewarded odor) increases while activity in the glomeruli/percepts which have a low uniqueness index (i.e., activated by both rewarded and habituated odors) decreases ($R^2 = 0.37$, $p = 0.0015$ for in vivo percepts and $R^2 = 0.76$, $p = 0.02$ for model percepts). Note, similar trend in vivo and in the model.

PP) resulted in a clear separation of odor representations in the coding space (Fig. 5C). Notably, both the correlations between odors from different classes (i.e., rewarded versus habituated) and those within the same class (e.g., PH1 vs. PH2, PP1 vs. PP2) decreased (Fig. 5B), aligning with our previous results obtained using the simpler model (Fig. 3B). This finding has also been observed in vivo[12], where a decrease in the similarity of representations was found between different PH and PP odors. This reduction in similarity indicates that differential conditioning might facilitate the recognition of individual rewarded odors.

Our minimal model (Fig. 3) predicts that neurons associated with percepts unique to the rewarded odors increase their firing rates after training, whereas neurons associated with percepts overlapping between rewarded and habituated odors reduce their activity. To quantify this effect in the more complex model (Fig. 5), we computed a uniqueness score for each percept as the normalized difference between its activation by the rewarded odor and its activation by the habituated odor (see "Methods"). Next, we calculated the change in each percept's activity following training (differential conditioning). The results, plotted in Fig. 5D (red dots and line), revealed that percepts with higher uniqueness indices increased their activity after training (indicating a positive correlation), while those with lower uniqueness indices decreased, thereby supporting our hypothesis.

To test the model's prediction, we sought to quantify this effect in vivo. The Ca²⁺ imaging data from honeybees was published previously[12], and here

we re-analyzed it. Similar to the model analysis, we calculated a uniqueness score for each glomerulus. This was done by determining, for each glomerulus, the difference between its activation by the rewarded odor and its activation by the habituated one (see "Methods"). We compared odor representations obtained after differential conditioning to those obtained after absolute conditioning, as experimental recordings for the naive condition are not available. We found a strong correlation between the uniqueness score of a glomerulus and the change in its activity due to differential conditioning compared to absolute conditioning (Fig. 5D, black dots and line). Specifically, glomeruli tended to increase their activity after training if they were unique in the representation of the rewarded odor, and vice versa, in agreement with our model's predictions. We term this effect, in which stimuli are positioned further apart in the coding space in accordance with the uniqueness of the percepts, *contrast enhancement*.

### The inhibitory AL network adapts to discriminate between odors in a new environment

As honeybees move from one environment to another, the odors associated with reward and habituation may change. The animal needs to learn the association of these new odors with the reward contexts present in the new environment. Changes in nectar production throughout the day and potentially many times within a forager's lifetime add to the complexity. The inhibitory AL network, therefore, must be flexible enough to learn

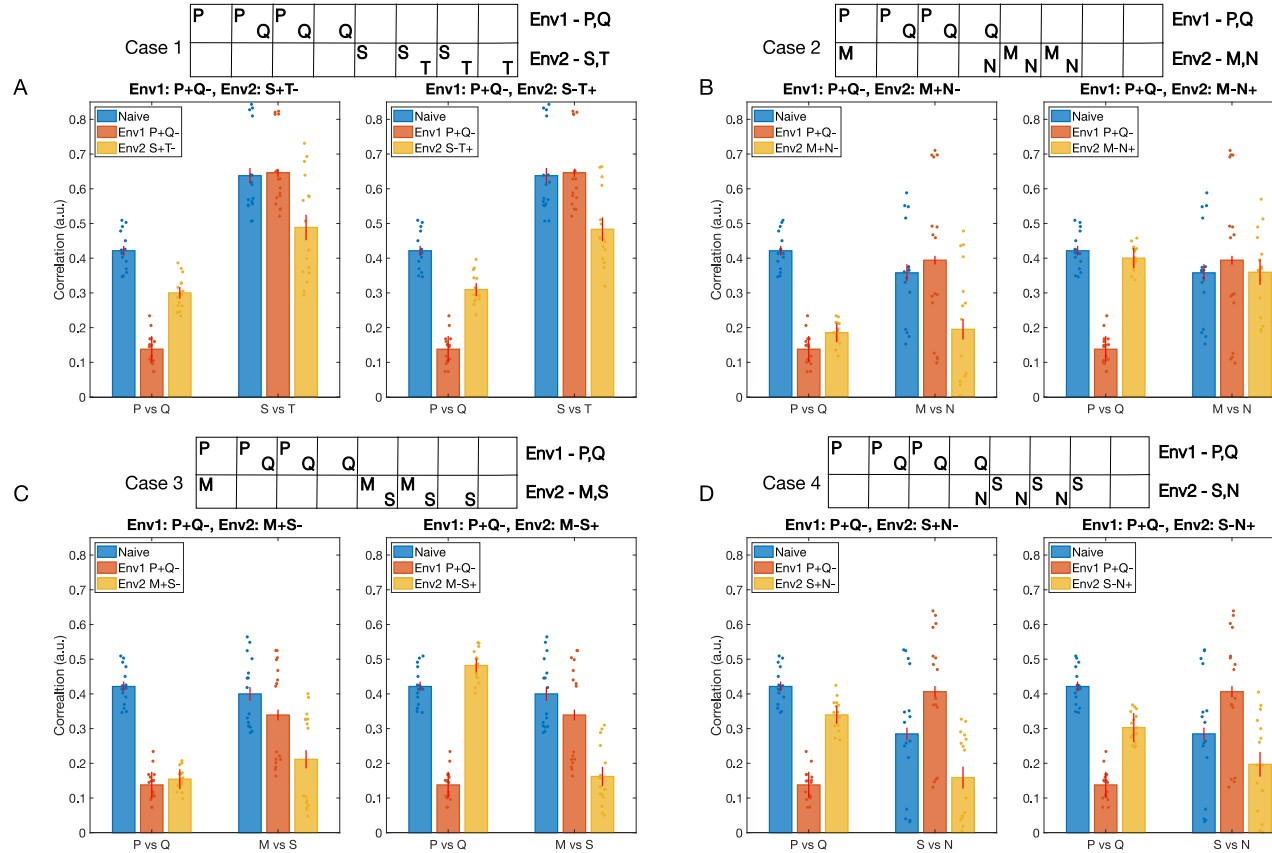

**Fig. 6 | Learning by the AL inhibitory network in different environments.** The network is initially trained in Environment 1 (Env1), which contains a rewarded odor class (P) and a habituated odor class (Q). It is then trained in a new Environment 2 (Env2) containing different odor classes. Bars represent the correlation between odor representations for odor classes P and Q from Env1 (left set of bars in each subplot) and for odor classes from Env2 (right set of bars) at three stages: naive (blue), after training in Env1 (orange), and after subsequent training in Env2 (yellow). Each **A**–**D** illustrates a different scenario of odor overlap between Env1 and Env2, with subpanels (left and right for each case) depicting different reward structures in Env2. **A** Case 1: No odor overlap between Env1 (odors: P, Q) and Env2 (odors: S, T). **B** Case 2: Complete overlap—both odors from Env2 (M, N) overlap with odors from Env1 (P, Q). **C** Case 3: Partial overlap—one odor from Env2 (M) overlaps with the rewarded odor from Env1 (P), while the second odor (S) does not overlap. **D** Case 4: Partial overlap—one odor from Env2 (N) overlaps with the habituated odor from Env1 (Q), while the second odor (S) does not overlap. The boxes at the top illustrate the distinct percepts activated by the odors in each environment, with overlapping percepts indicated by shared columns. Lower correlation values indicate better separation between odor representations. Error bars show the standard deviation across $n = 5$ trials. Dots represent individual data points. Significance values for each comparison are provided in Table 1.

representations for the new odors and successfully discriminate between them without immediately forgetting the old environment.

The odors from the new environment (we will call it Env2) may or may not overlap with the odors from the first environment (Env1) the animal has learned already. Below, we discuss four cases based on the overlap between the odors of the two environments. As before, each environment has two classes of odors (e.g., class P—rewarded, and class Q—habituated), each class containing 10 odors.

Case 1: the odor classes from Env1 (P and Q) do not overlap with the odor classes from Env2 (S and T). Case 2: both rewarded and habituated odor classes from Env1 and Env2 overlap. Here, Env2 consists of odor class M (overlaps with class P) and odor class N (overlaps with class Q). Case 3: one odor class from Env2 (class M) overlaps with the rewarded odor class from Env1 (class P), and the second odor class from Env2 (class S) does not overlap with any odors from Env1. Case 4: one odor class from Env2 (class N) overlaps with the habituated odor class from Env1 (class Q), and the second odor class from Env2 (Class S) does not overlap with any odor from Env1.

The structure of each odor class used in the experiments is shown in Fig. 6, at the top of the plots. Each class was based on three percepts, and, as before, individual odors within each class were obtained by varying the activation of individual percepts. The reward structure for the first environment is always such that odor class P is rewarded and class Q is habituated

(P+Q−). Each case is further divided into two sub-cases based on which odor class from the second environment is rewarded and which is habituated. Finally, we included a mechanism where synaptic weights return to baseline in the absence of reinforcement (see "Methods"). This "forgetting" mechanism was slow, so the network does not fully recover to its original naive state between training on Env1 and Env2. Because of this, in some cases, the effects of training on Env1 and Env2 were asymmetric, as Env2 was applied to a network that had already been pretrained on Env1.

Case 1: when there is no overlap between odors from Env1 and Env2, the correlation between representations of classes P and Q decreases significantly after training on Env1 (P+Q−), but increases again after training on Env2. This occurs because the weights associated with discriminating between P and Q odors receive no reinforcement during training on Env2 and slowly return to baseline. The correlation between odors S and T only decreases after training on Env2, regardless of the reward structure (S+T− or S−T+), indicating effective learning in the new environment without interference from prior learning (Fig. 6A).

Case 2: when both rewarded and habituated odors overlap between environments (Fig. 6B), we observe that if the reward structure remains consistent (M+N−) (Fig. 6B, left), the correlation between P and Q representations remains low after training on Env2. This is because training with overlapping Env2 promotes some of the same synaptic changes learned in Env1. However, when reward associations are reversed (M−N+)

**Table 1 | Statistical analysis results for odor discrimination**

| Case | Comparison | $p$ value | Case | Comparison | $p$ value |
|---|---|---|---|---|---|
| Case 1 pos | P/Q (Naive vs. Env1) | $1.44 \times 10^{-4}$*** | Case 3 pos | P/Q (Naive vs. Env1) | $1.44 \times 10^{-4}$*** |
| | P/Q (Env1 vs. Env2) | 0.002** | | P/Q (Env1 vs. Env2) | 0.423 |
| | P/Q (Naive vs. Env2) | 0.018* | | P/Q (Naive vs. Env2) | $1.21 \times 10^{-4}$*** |
| | S/T (Naive vs. Env1) | 0.931 | | M/S (Naive vs. Env1) | 0.557 |
| | S/T (Env1 vs. Env2) | 0.174 | | M/S (Env1 vs. Env2) | 0.292 |
| | S/T (Naive vs. Env2) | 0.206 | | M/S (Naive vs. Env2) | 0.089 |
| Case 1 neg | P/Q (Naive vs. Env1) | $1.44 \times 10^{-4}$*** | Case 3 neg | P/Q (Naive vs. Env1) | $1.44 \times 10^{-4}$*** |
| | P/Q (Env1 vs. Env2) | $8.29 \times 10^{-4}$*** | | P/Q (Env1 vs. Env2) | $1.94 \times 10^{-5}$*** |
| | P/Q (Naive vs. Env2) | 0.022* | | P/Q (Naive vs. Env2) | 0.152 |
| | S/T (Naive vs. Env1) | 0.931 | | M/S (Naive vs. Env1) | 0.557 |
| | S/T (Env1 vs. Env2) | 0.099 | | M/S (Env1 vs. Env2) | 0.105 |
| | S/T (Naive vs. Env2) | 0.128 | | M/S (Naive vs. Env2) | 0.016* |
| Case 2 pos | P/Q (Naive vs. Env1) | $1.44 \times 10^{-4}$*** | Case 4 pos | P/Q (Naive vs. Env1) | $1.44 \times 10^{-4}$*** |
| | P/Q (Env1 vs. Env2) | 0.062 | | P/Q (Env1 vs. Env2) | $2.77 \times 10^{-4}$*** |
| | P/Q (Naive vs. Env2) | $2.76 \times 10^{-4}$*** | | P/Q (Naive vs. Env2) | 0.061 |
| | M/N (Naive vs. Env1) | 0.813 | | S/N (Naive vs. Env1) | 0.415 |
| | M/N (Env1 vs. Env2) | 0.259 | | S/N (Env1 vs. Env2) | 0.088 |
| | M/N (Naive vs. Env2) | 0.240 | | S/N (Naive vs. Env2) | 0.334 |
| Case 2 neg | P/Q (Naive vs. Env1) | $1.44 \times 10^{-4}$*** | Case 4 neg | P/Q (Naive vs. Env1) | $1.44 \times 10^{-4}$*** |
| | P/Q (Env1 vs. Env2) | $1.49 \times 10^{-5}$*** | | P/Q (Env1 vs. Env2) | $1.80 \times 10^{-4}$*** |
| | P/Q (Naive vs. Env2) | 0.522 | | P/Q (Naive vs. Env2) | 0.009** |
| | M/N (Naive vs. Env1) | 0.813 | | S/N (Naive vs. Env1) | 0.415 |
| | M/N (Env1 vs. Env2) | 0.812 | | S/N (Env1 vs. Env2) | 0.141 |
| | M/N (Naive vs. Env2) | 0.988 | | S/N (Naive vs. Env2) | 0.498 |

*$p < 0.05$; **$p < 0.01$; ***$p < 0.001$. All $p$ values were calculated using unpaired two-tailed Student's $t$ tests.

(Fig. 6B, right), the decorrelation between P and Q increases again to the naive level, more so than in the case of non-overlapping odors, suggesting interference between the two environments.

Case 3: when only the rewarded odors overlap (class M overlaps with class P), the correlation between P and Q remains low when the reward structure of Env2 is consistent with Env1 (Fig. 6C, left), but it significantly increases when the reward structure of Env2 is opposite to Env1 (Fig. 6C, right). The network is able to learn new associations between M and S regardless of the reward structure of Env2 (Fig. 6C).

Case 4: when odors from Env2 (class N) only overlap with the habituated odors from Env1 (class Q), we observe an increase in correlation between P and Q after training on Env2, regardless of its reward structure (Fig. 6D). The network is also able to learn new associations between S and N. This suggests that changing the reward association of previously habituated components has minimal effect on existing memories compared to cases where rewarded components change their associations.

These results demonstrate that the AL network can effectively learn new odor-reward associations while maintaining existing ones, particularly when there is minimal overlap between environments or when reward structures remain consistent. The network shows more flexibility in adapting to new reward associations when they do not directly conflict with strongly learned rewarded odor components from the previous environment.

Notably, the overlap between an odor from Env2 and the rewarded odor from Env1 has a more significant impact on the odor correlations compared to an overlap with the habituated odor from Env1. This observation suggests that the network is more sensitive to changes in the reward associations of previously rewarded odors than to changes in the associations of previously habituated odors.

Our study predicts that it takes longer for the network to learn the second environment when the reward structure is reversed, which is consistent with experimental studies on reversal learning[39,40]. Latent inhibition refers to the observation that prior exposure to a stimulus without any associated reward can slow down the subsequent learning of associations involving that stimulus[41,42]. In the context of this experiment, the prior learning of reward associations in Env1 can interfere with the learning of new associations in Env2 when the odors overlap but have opposite reward structures, resulting in slower learning and adaptation to the new environment.

## Graph convolution neural networks and inhibitory learning converge on similar solutions

Can the principles learned from olfaction be exploited to improve performance of artificial neural networks (ANNs)? The encoding of odors in the AL takes place by learning the relationships between the components of the odors and the reward associated with the odors. Graph convolutional network (GCN) is a class of ANNs that take graph data as the input and aggregates information from neighboring nodes of the graph to perform a task (e.g., classification)[43,44]. This enables the GCN to utilize the relationships between nodes to improve the classification performance[44]. Hence, GCNs can be considered analogous to the AL, in that they encode relationships between different components of the input and utilize these relationships to categorize the inputs.

We compared results obtained using our biophysical model of the honeybee AL network to a GCN. To enable classification, the GCN was followed by a simple fully connected network, analogous to subsequent processing centers in the olfactory system (e.g., the mushroom bodies). We trained each network on chemical gas sensor data[45]: the biophysical network model was trained via inhibitory plasticity (as described previously) (Fig. 7), and the GCN was trained using backpropagation (see "Methods") (Fig. 8). The numerical features in this dataset—including response magnitude, On and Off time constants, among others (see

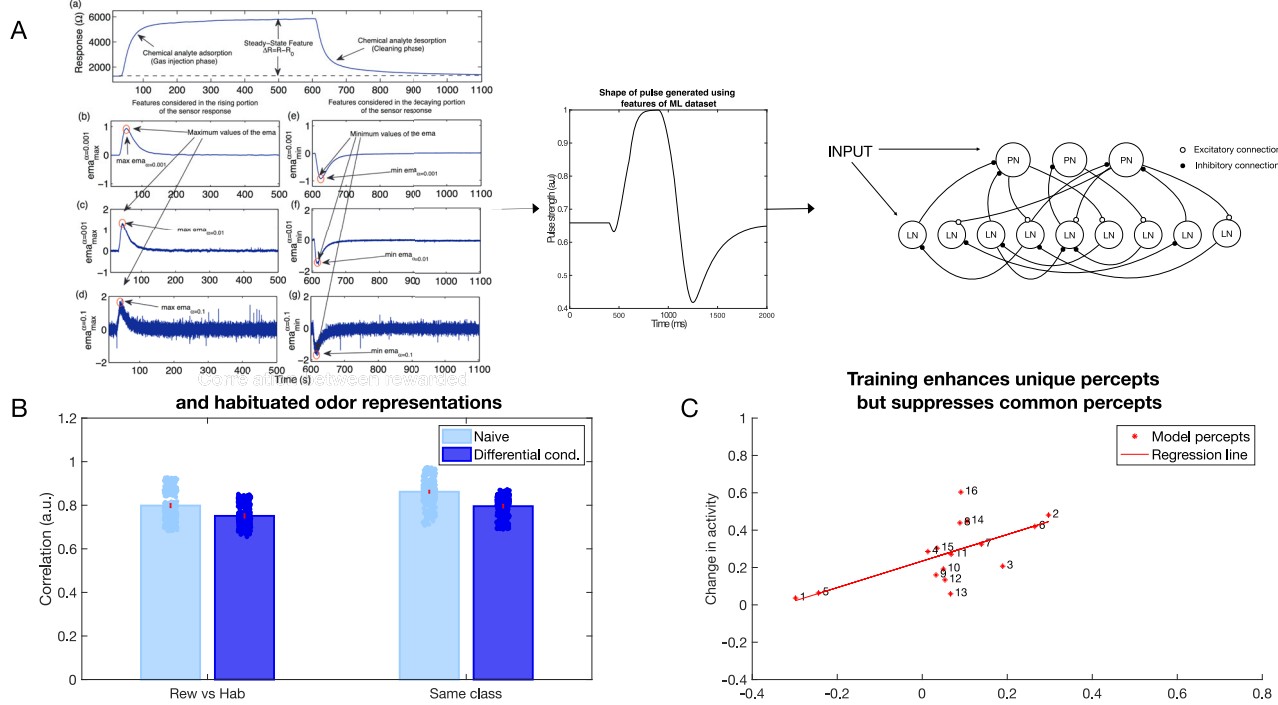

**Fig. 7 | Contrast enhancement in a large network model with inputs generated from chemical gas sensor data (Gas Sensor Array Drift Dataset from ref. 45).** **A** The features extracted from the response curves are converted into a pulse that is then fed into the larger biophysical network model as an input. **B** Correlation between (left set of plots, $p = 7.4e{-}18$) and within classes (right set of plots, $p = 1.6e{-}17$) decreases after differential conditioning (dark blue) compared to naive (light blue). The error bars represent standard deviation across $n = 10$ trials. Dots represent individual data points. **C** Contrast enhancement analysis for the large biophysical network model using chemical gas sensor data revealed similar results to the smaller biophysical network model and in vivo data ($R^2 = 0.416$, $p = 0.007$) (compare to Fig. 5D).

"Methods")—were used directly to train the GCN model. In the biophysical model, we used three features (response magnitude and On and Off time constants) from the data to construct input pulses that model odor stimulation (Fig. 7A).

We selected half of the odors to be associated with the reward and the other half to be habituated. Learning in the biophysical model increased the discriminability between the rewarded and habituated odors as indicated by the decrease in correlation (Fig. 7B). Note that this discrimination task was more difficult compared to the previous simulations, as identical sets of neurons were activated by different odors. Only the amplitude and On/Off time constants of the input were varied for each percept.

The GCN network (Fig. 8A) identified the class an odor belonged to (i.e., rewarded or habituated) with 88% accuracy. Importantly, both the GCN and the biophysical model exhibited similar changes in odor representations. Specifically, units representing components unique to an odor were enhanced, while units representing components common to many odors were suppressed (Figs. 7C and 8B). Both the top-down approach, based on backpropagation, and the bottom-up approach, grounded in local biological learning rules, converged on the same contrast enhancement strategy. These results underscore the effectiveness of contrast enhancement as a computational strategy for categorization tasks.

## Discussion

In vivo recordings from the honeybee's antennal lobe (AL) have shown that separation among patterns of neural activity representing odor blends from different varieties of flowers improves following differential reinforcement for each variety[12]. However, the mechanisms underlying this phenomenon remain poorly understood. The primary goal of this study was to uncover the plasticity mechanisms that modify neural representations of natural odor blends in the early olfactory system—the honeybee AL—using a combination of computational modeling, machine learning, and Ca$^{2+}$ imaging.

Natural odors are generally blends of several chemical components[1]. These odors occur in varying concentrations, and nectar production may vary from season to season and environment to environment. Although the olfactory coding space is extensive, optimally mapping the sensory environment requires learning and relearning the associations of rewards with variable blends of volatile compounds[1]. We found that, through a combination of associative and nonassociative plasticity in the inhibitory AL network, the activity of components common to both the rewarded and habituated odors decreases, while the activity of components unique to the rewarded odors increases after differential conditioning. This leads to a decreased overlap between the representations of the odor classes, while simultaneously making the representation of the rewarded odors more compact. This "contrast enhancement" effect was predicted by our biophysical computational model, and the model's predictions were confirmed by the Ca$^{2+}$ imaging data from the honeybee AL.

We also found that, upon subsequent stimulation with odors from a different environment, the inhibitory AL network learned the new odor-reward associations while preserving those established in the old environment. To test the application of these principles to Artificial Intelligence, we developed and trained a machine learning model based on the GCN to perform the same task of categorizing complex odors. By analyzing the activity of the units in the GCN, we observed a similar strategy of changes in the representation of rewarded odors as seen in the biophysical computational model and in vivo data.

Previous works have successfully applied computational network models to study the mechanisms of odor-induced synchronization[36–38,46,47] and neuronal plasticity[9,18,21,42,48–50] in the early olfactory system of insects. The study by ref. 9 used both firing rate and conductance-based models to explore nonassociative plasticity in the AL. The model in ref. 21 demonstrated that stimulus-specific changes in synaptic inhibition are sufficient to explain shifts in odor representations following olfactory learning.

**Fig. 8 | Contrast enhancement in a machine learning model. A** Schematic of the GCN model. The model receives features from the Gas Sensor Array Drift Dataset and processes it through two graph convolutional layers. The graph is then unrolled into a vector and sent through two fully connected layers for categorization. Training weights between GCN layers can be thought of as analogous to the biophysical model training. **B** Change in activity in the graph nodes in the first and second layer of the GCN vs. uniqueness index of that node ($R^2 = 0.276$, $p = 0.031$). Note contrast enhancement: nodes with higher uniqueness tend to increase the activity and those with lower uniqueness index tend to decrease the activity.

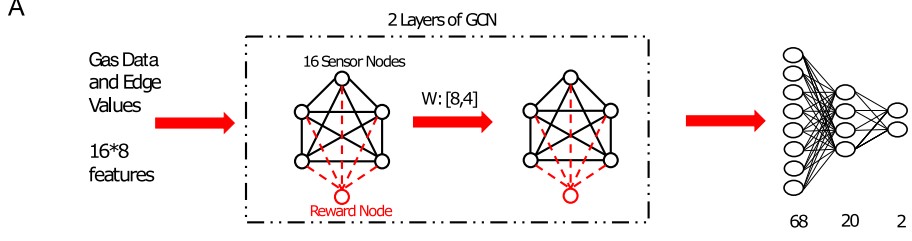

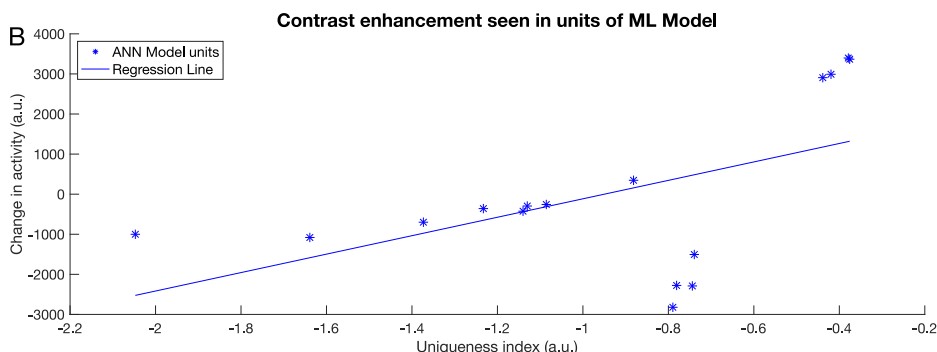

Our new study predicts that the AL network may utilize inhibitory synaptic plasticity to optimize its responses to odors associated with reward within a given odor environment. Considering the limited neuronal resources of the insect olfactory system and the complexity of the tasks it must solve, this study proposes a novel strategy by which the early olfactory system facilitates odor processing at downstream levels, focusing on odors relevant for survival. The model predictions align with imaging studies in the honeybee[12,51–53] and other insects[54] showing that odor representations in the AL change after differential conditioning, with rewarded and unrewarded odor representations becoming less correlated. These findings support the notion that the increase in separation of odor representations in the AL in vivo occurs through contrast enhancement, thereby validating the predictions of our model. Moreover, the main principles of nonassociative and associative plasticity operating in the first relay for olfactory coding appear to be conserved in the brains of both insects and vertebrates[8–12], suggesting that our model predictions can be generalized beyond insect olfaction.

Honeybees have a rich repertoire of learning behaviors to odors that range from nonassociative to associative and operant conditioning[3,55]. Biogenic amines have been shown to play an important role in reinforcement during olfactory learning[25,56,57]. In the honeybee, a cluster of cells at the base of the brain called Ventral Unpaired Medial (VUM) cells receive input from sucrose (US)-sensitive taste receptors[25,58–61]. The outputs from VUM cells spread broadly throughout association areas of the brain, including the AL and the MB, where they release the biogenic amines octopamine and tyramine as the reinforcement signal. Immunological studies have shown that the octopamine receptor (AmOA1) is expressed alongside GABA receptors in the honeybee AL[24]. Other studies[26] have shown that octopamine modulates odor representations in PNs by regulating inhibitory connections in the AL. While other types of octopamine receptors, such as Octβ1R and Octβ2R, are also expressed in the honeybee brain, here we focus on AmOA1[27], which is orthologous to Drosophila's DmOAMB receptor. In honeybees[28] and fruit flies[29], octopamine binding to this receptor leads to an increase in intracellular calcium, presumably resulting in an excitatory effect. Importantly, this receptor is present on a subset (i.e., not all) of the inhibitory interneurons in the antennal lobe of honeybees and fruit flies[27]. Thus, activation of these receptors on a subset of LNs may inhibit other LNs and thereby lead to partial PN disinhibition, as reported in ref. 26. Repeated or prolonged exposure to a stimulus without any reinforcement reduces the behavioral response to those odors, a process known as habituation[41,42,62], which leads to latent inhibition[41,63]. Studies in the fruit fly AL have revealed that habituation arises from the facilitation of inhibitory LN to PN synapses involved in odor representation[34,35].

The model predicts that both associative and nonassociative plasticity (habituation) are required to explain in vivo data. The suppression of the common component in the rewarded odors was driven by an increase in the inhibitory LN to PN connection strength, caused by postsynaptic (nonassociative) plasticity. This effect resulted from the firing of PNs triggered by the habituated odor, aligning with the findings of refs. 9, 21. These studies demonstrated that nonassociative inhibitory plasticity from LNs to PNs plays a crucial role in "filtering out" components of the habituated odor after honeybee was exposed to the odor without receiving a reward. The increase in the activity of the unique component in the rewarded odors occurred due to both presynaptic and postsynaptic facilitation, leading to an increase in inhibition of the LNs that were inhibiting PNs representing the unique components (i.e., PNs' disinhibition). A similar effect, an increase in the activity of the glomeruli for the rewarded but not the habituated odor, was also observed in ref. 51. Together, these two mechanisms contributed to shifting the PN responses for the rewarded odor away from those of the habituated odor. Our AL model focuses on the synaptic connections between excitatory PNs and inhibitory LNs. While those represent the canonical AL circuit, other types of connections are found in different insect species, e.g., in fly olfactory system the vast majority of LN also synapses onto ORNs[11]. While we acknowledge this model limitation, our focus on inhibitory plasticity (LN-to-PN and LN-to-LN synapses) reflects our central hypothesis that experience-dependent modifications of inhibitory circuits reshape odor representations in the AL.

The mushroom body (MB), the next layer in olfactory processing in insects, is considered a major center for associative learning and olfactory memory[56]. Information from the PNs in the AL is relayed to the MB, where Kenyon cells (KCs) represent odors with sparse firing patterns[64,65]. Remarkably, approximately 800 PNs synapse onto around 170,000 KCs[66,67], significantly increasing the dimensionality of the odor representation space. Experimental studies have shown that KC responses are influenced by associative reinforcement learning, which stabilizes odor representations in the MB. In contrast, odor presentations without reinforcement weaken their representation in the MB[68]. Modeling studies[69] have also found that the categorization of odors in the MB depends on changes in PN responses caused by conditioning. Our study suggests a potential mechanism for this change in KC responses. After training, the rewarded odor triggers more population-specific AL activity, leading to more reliable firing of the KCs receiving input from these PNs. This observation aligns with the findings of ref. 68.

Artificial intelligence (AI) has made rapid progress in recent years in solving tasks such as image classification, speech recognition and natural language processing. Although it has now grown into a separate field, AI can trace its roots back to neuroscience with the earliest artificial neural networks designed to mimic information processing in the brain[70]. Currently, efforts are being made to incorporate biological principles to develop a new generation of AI[71,72]. Graph Neural Networks (GNNs) have been developed to work with graph data. There have been significant advances in GNNs, increasing their capabilities and expressive power[44,73]. In one study, GNNs were used to learn a generalizable perceptual representation of odors[74]. Here, we developed a graph convolutional network (GCN) model as an artificial parallel to the insect antennal lobe. We trained this network to perform an odor categorization task and analyzed the activity of the artificial neurons in the first and second layers of the GCN. This analysis revealed that the contrast between representations of different odor classes increased via the same mechanism described in the biophysical computational model: units representing components unique to an odor were enhanced, while units representing components common to many odors were suppressed. Together, these results suggest that contrast enhancement is an efficient strategy for performing a categorization task.

In summary, our study has uncovered novel circuit-level mechanisms of olfactory learning in the honeybee AL that enhance odor discrimination. This learning paradigm relies on inhibitory network plasticity, which allows the coding space to be reshaped to align with the current task and environment. These findings suggest an effective computational strategy for the perceptual learning of complex natural odors, achieved through changes in the inhibitory network during early sensory processing.

## Methods
### Animals and calcium imaging experiments
Pollen forager honey bees (*Apis mellifera*) were collected from regular hives and restrained in individual harnesses suited for olfactory conditioning and calcium imaging recordings. Appetitive olfactory conditioning followed standard protocols[75]. Two groups of bees received different conditioning protocols using odor blends that mimicked the natural odor variation within and between two varieties of Snapdragon flowers, Potomac Pink (PP) and Pale Hybrid (PH)[12,22]. One group of bees was subject to differential conditioning (PH+/PP−) by rewarding with 1M sucrose solution odor blends from the PH variety and not rewarding blends of the PP variety. The second group of bees (PH+/MO−) was trained rewarding the same PH blends as for the first group, but the odor solvent mineral oil (MO) was used instead PP blends for unrewarded trials. Thus, the difference between groups was in the unrewarded trials. Eight hours after conditioning, uniglomerular projections neurons of the antennal lobe were stained by backfilling with the calcium sensor dye fura-dextran (potassium salt, 10.000 MW, Thermo Fisher Scientific)[52]. Calcium imaging was performed on the next day. The head-capsule was opened and rinsed with Ringer solution (130 mM NaCl, 6 mM KCl, 4 mM MgCl2, 5 mM CaCl2, 160 mM sucrose, 25 mM glucose, 10 mM HEPES, pH 6.7, 500mOsmol; all from Sigma-Aldrich). Glands and trachea covering the antennal lobes were removed. Only one antennal lobe per bee was used for calcium imaging. Calcium imaging was done using a CCD camera (SensiCamQE, T.I.L.L. Photonics) mounted on an upright fluorescence microscope (Olympus BX-50WI, Japan) equipped with a 20x objective, NA 0.95 (Olympus), 505 DRLPXR dichroic mirror, and 515 nm LP filter (TILL Photonics). Monochromatic excitation light provided by a Polichrome V (TILL Photonics) alternated between 340 nm and 380 nm. Images were recorded at a sampling rate of 8 Hz. Spatial resolution was $172 \times 130$ pixels, after a binning of $8 \times 8$ on a chip of $1376 \times 1040$ pixels, resulting in a spatial sampling of 2.6 µm per pixel side. Each data set consisted in a double sequence of 80 images, obtained at 340 nm and 380 nm excitation light respectively (Fi340, Fi380, where i is the number of images from 1 to 80). Calcium signals were calculated using a ratiometric method: for each pair of images Fi we calculated $R_i = (F_i 340/F_i 380) \times 100$ and subtracted the background Rb that was obtained by averaging the Ri values 1s before odor onset. The analysis was based on the calcium signals from the same 24 glomeruli

identified in all bees (Fig. 1C)[76]. The glomerular activation was considered as representing the activity of uniglomerular PNs, as only these neurons in the AL were stained. The odor delivery device consisted of a main charcoal filtered air stream (500 ml/min) and fifteen identical odor channels, each composed of one valve and one odor cartridge. The odor cartridges consisted of a 1ml syringe containing a paper strip loaded with 10 ul odor solution. A three-way valve (LFAA1200118H; The LEE Company) controlled the airflow through the odor cartridge. Opening of the valves was synchronized with the optical recordings by the acquisition software TILLVisION (TILL Photonics). During stimulation with odor, 4 ml of air loaded with the odor was added to the main air stream during 4 s. Calcium imaging analysis was designed to determine how both trainings protocols affect the neural representation of PH and PP blends. The calcium activity measured between 375 ms and 625 ms after odor onset was averaged, thus the activation pattern elicited by each odor was reduced to a single vector of 24 elements (glomeruli) (see Fig. 1A for examples). This time interval includes the time points in which odor representations reach the minimal correlation after odor onset. The similarity between 2 glomerular odor patterns was calculated as the correlation between the respective 24-dimensional vectors. As we measured glomerular activity patterns elicited by 6 PH blends and 6 PP blends, we could calculate for each animal the average distance between 36 pairs of PH-PP blends. This was done for 10 absolute conditioned bees and 7 differentially conditioned bees. This data has been previously published in ref. 12. In this manuscript, we re-analyze this calcium imaging data.

### Electrophysiology data
To perform extracellular electrophysiology, a Tucker-Davis Technologies (TDT) RZ2 microprocessor system, in conjunction with a PZ2 preamplifier from the same manufacturer (TDT®, Alachua, FL 32615 USA) was used to digitize neural signals, which was sampled with a Neuronexus A2 × 2 multichannel probe (Neuronexus®, Ann Arbor, MI 48108 USA), as described in ref. 77. Briefly, the probe was carefully inserted into the central neuropil of the AL aided with a Leica micromanipulator. After initial contact, the probe was slowly moved deeper, a few microns each step, until spikes appeared. Care was taken to insert probes approximately in the same location each time. The acquisition software (Synapse©TDT) was configured to acquire the spike waveforms at 25 kHz sampling rate, and the 16 recording channels were grouped in the software to form 4 tetrodes[78]. To quantify the neural responses, the tetrode waveforms, which are concatenated from four the waveforms simultaneously captured at the four contact sites within a tetrode, were exported from the TDT acquisition software to the Offline Sorter program (Plexon®Inc., Dallas, TX USA), which allows automatic as well as manual sorting of the waveforms. The quality of spike sorting was statistically verified within Offline Sorter. The time stamps of all waveforms were then exported to a spike analysis program, Neuroexplorer ®(Nex Technologies, Dallas, TX USA) or Matlab ®(Mathworks, Natick, MA USA) for further analysis. Units were further classified into PNs and LNs using the method outlined in refs. 79,80. In short, PNs and LNs in honeybee AL differ in their spiking properties, including changes in firing rate due to stimulation, latency to response, variability in interspike intervals and variability of responses to odor pulses. These features were then used in a hierarchical clustering analysis using the Ward linkage method and the Euclidean distance metric (calculations were carried out with scipy's linkage and fcluster functions) to separate putative PNs and LNs.

### Computational modeling
We constructed a biophysical computational model of the honeybee antennal lobe (AL), containing 100 excitatory projection neurons (PNs) and 280 inhibitory local interneurons (LNs). Each PN and LN was modeled by a single compartment that included voltage- and calcium-dependent currents described by Hodgkin-Huxley kinetics[81]. The model here was adapted from our previous studies[21,82] and is available for download from the Bazhenov lab website. Model equations were solved using a fourth order Runge-Kutta method with an integration time step of 0.04 ms.

**Membrane potentials.** PN and LN membrane potential equations[81] are given by:

$$C_m \frac{dV_{PN}}{dt} = - g_L(V_{PN} - E_L) - I_{Na} - I_K - I_A - I_T - I_h - g_{KL}$$
$$(V_{PN} - E_{KL}) - I_{GABA_A} - I_{\text{slow-inh}} - I_{nACh} - I_{stim}$$

$$C_m \frac{dV_{LN}}{dt} = - g_L(V_{LN} - E_L) - I_{Na} - I_K - I_T - g_{KL}(V_{LN} - E_{KL})$$
$$- I_{GABA_A} - I_{nACh} - I_{stim}$$

The passive parameters are as follows: For PNs, $C_m = 2.9*10^{-4}\,\mu F$, $g_L = 0.01\,mS/cm^2$, $g_{KL} = 0.012\,mS/cm^2$, $E_L = -70\,mV$, $E_{KL} = -95\,mV$.

For LNs, $C_m = 1.43*10^{-4}\,\mu F$, $g_L = 0.05\,mS/cm^2$, $g_{KL} = 0.018\,mS/cm^2$, $E_L = -70\,mV$, $E_{KL} = -95\,mV$. An external DC input was introduced to each neuron through $I_{stim}$.

The intrinsic currents, including a fast sodium current, $I_{Na}$, a fast potassium current, $I_K$, a transient potassium A-current, $I_A$, a low threshold transient $Ca^{2+}$ current, $I_T$, a hyperpolarization activated cation current $I_h$ are given by equation:

$$I = gm^M h^N(V - E)$$

where the maximal conductances for PNs are $g_{Na} = 90\,mS/cm^2$, $g_K = 10\,mS/cm^2$, $g_A = 10\,mS/cm^2$, $g_T = 2\,mS/cm^2$, $g_h = 0.02\,mS/cm^2$. The maximal conductances for LNs are $g_{Na} = 100\,mS/cm^2$, $g_K = 10\,mS/cm^2$, and $g_T = 1.75\,mS/cm^2$.

For all cells, $E_{Na} = 50\,mV$, $E_K = 95\,mV$, and $E_{Ca} = 140\,mV$. The gating variables $0 \le m(t), h(t) \le 1$ satisfy the following:

$$\frac{dm}{dt} = \frac{m_\infty(V) - m}{\tau_m(V)}$$

$$\frac{dh}{dt} = \frac{h_\infty(V) - h}{\tau_h(V)}$$

where the steady state (e.g., $m_\infty(V)$) and relaxation times (e.g., $\tau_h(V)$) are derived from experimental recordings of the specific ionic currents. These distinct voltage-dependent functions are given in ref. 21. For all cells, intracellular $Ca^{2+}$ dynamics were described by a simple first-order model as follows:

$$\frac{d[Ca^{2+}]}{dt} = -A \cdot I_T - \frac{[Ca^{2+}] - [Ca^{2+}]_\infty}{\tau}$$

where $[Ca^{2+}]_\infty = 2.4 \cdot 10^{-4}\,mM$ is the equilibrium of intracellular $Ca^{2+}$ concentration, $A = 5.2 \cdot 10^{-5}\,mM\,cm^2/(ms\,\mu A)$ and $\tau = 5\,ms$.

**Synaptic currents.** Fast GABA and cholinergic synaptic currents to LNs and PNs were modeled by first-order activation schemes[83]. Fast GABA and cholinergic synaptic currents are given by:

$$I_{syn} = g_{syn}[O](V - E_{syn})$$

where the reversal potential is $E_{nACh} = 0\,mV$ for cholinergic receptors and $E_{GABA} = -70\,mV$ for fast GABA receptors. The fraction of open channels, $[O]$, is calculated according to equation:

$$\frac{d[O]}{dt} = \alpha(1 - [O])[T] - \beta[O]$$

For the transmitter concentration of the cholinergic synapses, $[T]$, is given by:

$$[T] = A \cdot H(t_0 + t_{max} - t) \cdot H(t - t_0)$$

and for the GABAergic synapses:

$$[T] = \frac{1}{1 + \exp(-(V - V_0)/\sigma)},$$

where $H$ is the Heaviside step function, $t_0$ is the time of receptor activation, $A = 0.5$, $t_{max} = 0.3\,ms$, $V_0 = -20\,mV$ and $\sigma = 1.5$. The rate constants were given $\alpha = 10\,ms^{-1}$ and $\beta = 0.2\,ms^{-1}$ for fast:

$$GABA_A$$

synapses, and $\alpha = 1\,ms^{-1}$ and $\beta = 0.2\,ms^{-1}$ for cholinergic synapses.

The slow inhibitory synaptic current is given by the following equations:

$$I_{\text{slow-inh}} = g_{\text{slow-inh}} \frac{[G]^4}{[G]^4 + K}(V - E_k),$$

$$\frac{d[R]}{dt} = r_1(1 - [R])[T] - r_2[R],$$

$$\frac{d[G]}{dt} = r_3[R] - r_4[G]$$

where $[R]$ is the fraction of activated receptors, $[G]$ is the concentration of G proteins, $E_K = -95\,mV$ is the potassium reversal potential. The rate constants were $r_1 = 0.5\,mM^{-1}\,ms^{-1}$, $r_2 = 0.0013\,ms^{-1}$, $r_3 = 0.1\,ms^{-1}$, $r_4 = 0.033\,ms^{-1}$ and $K = 100\,\mu M^4$. The maximum synaptic conductances were set at $g_{GABA_A} = 0.02\,\mu S$ between LN, $g_{GABA_A} = 0.015\,\mu S$ from LN to PN, $g_{ACh} = 0.3\,\mu S$ from PN to LN and $g$slow-inh $= 0.02\,\mu$ from LN to PN.

For simulating the larger network with the gas sensor array drift dataset as input, the maximum synaptic conductances were as follows: $g_{GABA_A} = 0.024\,\mu S$ between LNs, $g_{GABA_A} = 0.019\,\mu S$ from LNs to PNs and $g_{ACh} = 0.075\,\mu S$ from PNs to LNs.

**Plasticity.** A simple phenomenological model of synaptic facilitation was used to model inhibitory plasticity (from LN to LN and LN to PN). Specifically, the inhibitory network underwent presynaptic facilitation when presented with a rewarded odor and postsynaptic facilitation when presented with a habituated odor during learning[21]. Presynaptic (associated with reward) facilitation was assumed to depend on the octopamine receptors AmOA1 expressed in inhibitory LNs in the honeybee AL[24,27] and it was based on the spiking events of the presynaptic neurons. In contrast, postsynaptic facilitation (associated with odor habituation) was based solely on the spiking event of the postsynaptic neuron[15,34].

To model facilitation, the maximum synaptic conductance is multiplied by a facilitation variable, $F$. $F$ is updated each time there is a spike:

$$F_{t+1} = F_t + dF_{pre} + dF_{post}$$

where $dF$ is facilitation rate ($dF_{pre} = 0.15$ and $dF_{post} = 0$ for presynaptic facilitation and $dF_{pre} = 0$ and $dF_{post} = 0.15$ for postsynaptic facilitation). $F_t$ is the value of the facilitation variable just before the spike. In the absence of any spiking events, $F$ had an exponential decay according to the following equation:

$$F_t = 1 + (F_{t-1} - 1) * \exp\frac{-((t - 1) - t_i)}{\tau}$$

**Table 2 | Connection probabilities between different neuron groups in the AL**

| Source unit | Target unit | Connection probability |
|---|---|---|
| All PNs | All PNs | 0 |
| | Unipolar LNs | 0.4 |
| | Multipolar LNs | 0.4 |
| Unipolar LNs | PNs from same glomerulus | 0 |
| | PNs from different glomerulus | 0.5 |
| | Unipolar LNs from same glomerulus | 0 |
| | Unipolar LNs from different glomerulus | 0.4 |
| | Multipolar LNs | 0.3 |
| Multipolar LNs | All PNs | 0.3 |
| | Multipolar LNs | 0.1 |
| | Unipolar LNs | 0.4 |

In the above equation, $\tau = 30$ s is the time constant of the decay; $t_i$ is the time of the $i$th spiking event; $F_t$ corresponds to the value of $F$ at time $t$ with the initial value $F_0 = 1$. The time constant of decay is the same for pre and postsynaptic plasticity. The updated synaptic weights at the end of the training period (30 presentations, 60,000 ms) were frozen and used as the synaptic weights in the testing phase (Fig. 1E).

**Network geometry.** The AL network included 100 PNs and 280 LNs. All these PNs and LNs were organized into 20 glomeruli, with each glomerulus containing 12 LNs (unipolar LNs) and 5 PNs. In addition to these, there are 40 LNs that contribute to global inhibition (multipolar LNs). The connections between each group of neurons were generated randomly based on the probabilities described in Table 2.

In order to accommodate the ML-dataset inputs to the model, in some simulations we increased the number of neurons to 400 PNs and 1120 LNs. The larger network has random connectivity between LNs, LNs to PNs and PNs to LNs with a probability of 0.125. There were no connections between the PNs.

**Odor stimulation.** To model odor stimulation, a fraction of LNs and PNs was activated by a current pulse with a rise time constant of 66.7 ms and a decay time constant of 200 ms for a period of 500 ms as can be seen in Fig. 2F. In addition, small amplitude current in the form of Gaussian Noise was added to each cell, to ensure random and independent membrane potential fluctuations. Each odor was modeled as a mixture of "pure" chemicals. Each pure chemical is defined here as a percept. Each percept activates a group of "virtual ORNs", which in turn activate a set of PNs. The odors we created activate 3 out of 7 percepts, which means each odor is a mixture of 3 "pure" chemicals out of a total of 7 "pure" chemicals.

For example, for simulations shown in (Fig. 3), the odor classes (A and B) were defined based on which 3 out of the 7 percepts were active for a given class. One odor class (A) was associated with reward and another one (B) with no reinforcement, with an overlap of 2 percepts. Individual odors within a class were defined by changing the level of LN/PN activation triggered by percepts.

In some simulations we modeled "PH-PP" odors which were created from the chemical compositions of two varieties of snapdragon odor blends (flowers)—PH (Pale Hybrid) and PP (Potomac Pink) as used in ref. 12. Since there are 6 chemical components present in each odor, each chemical component was assigned a percept and the level of activation of each percept (width of the gaussian centered around the center of the percept in Fig. 5A) was calculated based on the proportion of that chemical component present in the natural odor blend. The 7th percept remained inactive for all odors. For example, in the odor PH1, Oci chemical is 37.2% of the mixture. Hence,

the standard deviation, which determines the width for the Gaussian, for percept 1 is set to 0.372 (Fig. 5A).

Finally, we also used odor classes derived from the UCI Gas Sensor Array Drift dataset[45]. This dataset consists of data for 6 odors, obtained from 16 metal oxide sensors. Each odor consisted of 16 percepts, which activated a certain group of PNs and LNs. The first 200 out of 400 PNs and first 560 LNs out of 1120 LNs were provided with inputs, whereas the second half of PNs and LNs received no odor related inputs. This was done in order to maintain the E/I balance in the larger network. The first half of PNs and LNs were provided with odor pulses, created from 3 features of the data from each metal oxide sensor—$\Delta R$, $ema\_max_{\alpha=0.001}$ and $ema\_min_{\alpha=0.001}$[45]. From these 3 features, we created an odor pulse with a rise time constant proportional to $1/ema\_max_{\alpha=0.001}$, decay time constant proportional to $1/ema\_min_{\alpha=0.001}$ and the maximum height of the pulse proportional to $\Delta R$. Of the 6 odors, 3 were associated with reward and 3 were associated with no reinforcement.

**Machine learning model—GCN model.** In the last section of this paper, we applied a Graph Convolutional Network (GCN), a type of Graph Neural Network (GNN), to simulate AL training using machine learning approach. GCN is an approach for semi-supervised learning on graph structured data. It is convolutional in nature, because filter parameters are typically shared over all locations in the graph. GCN is a generalized version of CNNs which operate directly on arbitrarily structured graphs[43]. In ref. 44, the choice of convolutional architecture is motivated via a localized first-order approximation of spectral graph convolutions.

For the GCN model, the goal is to learn a function of signals/features on a graph which takes as input:

- A feature description $x_i$ for every node $i$; summarized in a N × D feature matrix X (N: number of nodes, D: number of input features).
- A representative description of the graph structure in matrix form; typically in the form of an adjacency matrix A (or some function thereof) and produces a node-level output Z (an N × F feature matrix, where F is the number of output features per node). Graph-level outputs can be modeled by introducing some form of pooling operation[43].

Every neural network layer can then be written as a non-linear function $H(l + 1) = f(H(l), A)$ with $H(0) = X$ and $H(L) = Z$, L being the number of layers. The specific models then differ only in how $f$ is chosen and parameterized.

The UCI Gas Sensor Array Drift Dataset was taken as the input to the GCN, which consists of 2 parts, i.e., the graph convolutional layers and the fully connected layers. This dataset contains data for 6 different odors from 16 metal oxide sensors, with 8 features extracted from each sensor response, as described above. Thus, each odor is made of 128 features from 16 sensors. The data was normalized by subtracting the mean and dividing by standard deviation for each sensor value over the training and test sets. The GCN model consists of 16 (sensor) + 1 (reward/habituation) nodes and 2 layers of graph convolution. The output of the GCN is converted to a vector and sent to 2 fully connected layers for the final classification. The 6 odors from the dataset are split into 2 groups, one group associated with reward and one group associated with habituation. The task being performed by the model is to determine which group the given odor belongs to. The equation of this GCN model is, then, $H(l + 1) = h(A \cdot H(l) \cdot W(l))$, where $l$ is the current layer and A is the graph adjacency matrix with all connections between nodes belonging to sensors set to 1. The reward/habituation nodes' connections to the main 16 node graph was set during training as follows: if a rewarded odor was presented to the model during training, the connections between the reward and sensor nodes was set to 1. Conversely, if a habituated odor was presented to the model during training, the connection between reward node and sensor nodes was set to −1.

The model was trained via backpropagation, with the matrix W and the fully connected head of the model being updated during training. The cross entropy loss was optimized with the Adam Optimizer with a learning rate of 0.01 for 20 epochs. The graph convolution part of the model takes in 8

features from the 16 sensors and transform them into 4 abstract ones at the second layer. These abstract features encode the relationships between the different sensor features during training, similar to the AL which learn the relationships between the different odor percepts during training. Hence, we can say that the GCN model performs input preprocessing by learning the relationships between input features similar to the AL in honeybees. The first layer of the GCN can be thought of as analogous to the untrained AL network and the second layer can be thought of as the trained AL network.

## Analysis

**Correlation analysis.** Correlation between odor representations was calculated based on the normalized binned spike counts across all the PNs for a single trial during the period of stimulation, with a bin size of 100 ms. The correlation was calculated between arrays of PN activity for each time bin and each trial and averaged over both time bins and trials.

**PCA trajectory.** The response trajectory of the entire PN population to a specific input was constructed by binning the spike trains in 40 ms bins. For visualization, the 100 dimensional space of PN responses was reduced to 2D/3D space using Principal Component Analysis (PCA).

**Uniqueness index calculation.** The uniqueness index for each glomerulus/percept (unit) was calculated as follows:

$$UI = \frac{\left(act\_unit_{naive}^{rew} - act\_unit_{naive}^{hab}\right)}{act\_unit_{naive}^{hab}}$$

and the change in activity of the neurons:

$$Change\ in\ activity = \frac{act\_unit_{diff}^{rew} - act\_unit_{naive}^{rew}}{act\_unit_{naive}^{rew}}$$

where $act\_unit_{naive}^{rew}$ denotes the activation of the glomerulus by the rewarded odor and $act\_unit_{naive}^{hab}$ denotes the activation by the habituated odor before training (in the naive condition) and $act\_unit_{diff}^{rew}$ denotes the activation of the glomerulus by the rewarded odor after differential conditioning. A high positive Uniqueness index (UI) indicates that the given glomerulus is uniquely activated by the rewarded odors and a UI close to 0 would indicate the glomerulus is activated to a common extent by both rewarded and habituated odors. This method of calculating the UI was used for the biophysical computational models.

The uniqueness index for the GCN model was calculated based on the average activation (average of 8 feature values) for each sensor in GCN layer 1 for rewarded and habituated odors. The UI for the GCN is:

$$UI = \frac{act\_unit_{L1}^{rew} - act\_unit_{L1}^{hab}}{act\_unit_{L1}^{hab}}$$

where $act\_unit_{L1}^{rew}$ corresponds to the average activation of the GCN unit for the rewarded odors in layer 1 and $act\_unit_{L1}^{hab}$ corresponds to the average activation of the GCN unit for the habituated odors in layer 1. The change in activity was calculated based on the difference in average activation of each sensor node between layer 1 and layer 2 of the GCN:

$$Change\ in\ activity = act\_unit_{L2}^{rew} - act\_unit_{L1}^{rew}$$

## Statistics and reproducibility

Statistical analysis was performed using unpaired two-tailed Student's *t* tests to compare the correlation between the different conditions for the computational model. The Wilcoxon signed rank test was used for the comparison of correlation between representations for the calcium imaging data in Fig. 1. All statistical tests were performed in MATLAB (MathWorks). Exact *p*-values are reported in the figure legends. For computational modeling experiments, n represents the number of independent simulation runs

with different random seeds, which we call trials. For the calcium imaging experiments, n represents the number of different bees used for each condition. The computational model code used in this study is available at Dr. Bazhenov's lab website as well as at https://doi.org/10.17605/OSF.IO/45TAY. All simulations are deterministic given the same random seed.

**Reporting summary**

Further information on research design is available in the Nature Portfolio Reporting Summary linked to this article.

## Data availability

The experimental data and the simulation results that support the findings of this study are available with the identifier: https://doi.org/10.17605/OSF.IO/45TAY[84].

## Code availability

The code for computational model simulations can be found at: https://github.com/bazhlab-ucsd/honeybee-learning. The code for generating figures and all the numerical data required to do so can be found at: https://doi.org/10.17605/OSF.IO/45TAY[84].

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

## Acknowledgements

This work was supported by grants—NSF (2223839, 2323241), NIH (R01DC020892), NSF/CIHR/DFG/FRQ/UKRI-MRC Next Generation Networks for Neuroscience Program (2014217), National Science Foundation CRCNS (2113179), and National Science Foundation NeuroNex Odor2Action (1559632).

## Author contributions

S.J., S.H., Z.W., M.B. and Y.C. conceived and designed the research. S.J., S.H., F.L., H.L. and Z.W. performed the experiments. S.J. and S.H. analyzed the data. S.J. and S.H. drafted the manuscript. S.J. prepared the figures. S.J., S.H., Z.W., F.L., H.L., B.S., Y.C. and M.B. edited and revised the manuscript.

## Competing interests

The authors declare no competing interests.
