## [Transparent Peer Review file · Communications Biology]

Plasticity in inhibitory networks improves pattern separation in early olfactory processing

Corresponding Author: Professor Maxim Bazhenov

Version 0:

Reviewer comments:

Reviewer #1

(Remarks to the Author)

In this manuscript Joshi et al. use computational modelling to try and demonstrate that associative and non-associative plasticity in the honeybee antennal lobe (AL) work together to enhance the contrast between rewarded and unrewarded odors.

In general, this manuscript has the potential to contribute to the field. However, there are some major concerns that reduce my enthusiasm for the current version of the manuscript.

Major comments:

1. The authors attempt to provide physiological evidence to the construction and execution of the model. However, in some cases these justifications are inaccurate, and, in some cases, physiological evidence is ignored.

For example, the authors present “odor” responses of PNs and LNs (Figure 2C, D). However, these responses are not similar at all to published recordings from the honeybee (See for example Meyer et al. 2013, Krofczik et al. 2009). From the presented figures, PNs and LNs have very low firing rates but in reality, at least PNs can reach very high firing frequencies. Similarly, PNs and LNs are modeled to have a rather step-like response, whereas, in reality, the response is highly dynamic.

Another point related to the model construction is the ORN input and its transformation to PNs and LNs. PNs and LNs are simulated in the same manner and with the same response profile. However, at least in flies, this was shown not to be the case. Rather, PNs have a non-linear response to ORN input whereas LNs have a linear response. Furthermore, LNs hardly respond to low odor concentration whereas PNs do.

I am not implying that every biological feature needs to be modeled, but rather that the model should have been more carefully and accurately constructed.

In honeybees there are five octopamine receptors. The authors cite a manuscript that showed that one type of octopamine receptor is expressed in GABAergic LNs and therefore they conclude that octopamine exerts its effect in presynaptic terminals of LNs. I find this a bit of a stretch.

Similarly, in flies Oct β 1r and Oct β 2R were reported to have opposite effects. Furthermore, it was suggested that Oct β 1r reduces cAMP levels whereas Oct β 2R increases cAMP levels (Koon and Budnik, JNS 2012). Finally, cAMP level by itself is not an indication to whether excitation or inhibition eventually occurs. How can one conclude that octopamine facilitates LN presynaptic activity? Also, *Amoa1* was reported to increase overall glomerular activity (Rein et al. 2013). The paper (Rein et al.) that is used to support the assumption that OA modulates LN presynaptic terminals actually states: “Rather, it is likely driven by external factors, e.g. by decreased inhibitory input” – thus they reach the opposite conclusion about OA modulation of LN activity.

To summarize this part, this is an interesting model, but it cannot be presented as a realistic model.

2. The authors do not show an exploration of the model parameter space. The model’s parameters seem to be very specific. How they were selected was not explained. More importantly, how sensitive are the conclusions to changes in the model’s

parameters was not demonstrated. Do slight changes in the parameters completely change the model outcomes? This could perhaps be negated in a very realistic model in which its parameters are completely based on measured results. However, as explained in point 1 above, this is not the case here.

3. The authors use a confusing approach regarding the matrices they are using. In Figure 1 they show the non-significant Euclidean distance result and not the significant correlation result that the authors mention they have. Later, they use the model to show significant Euclidean distance results. So, what are we to believe in? The physiological results that show no significant difference between absolute and conditional conditioning, the model results that do show a significant effect or the results that were not even presented in the manuscript. What can we learn from a model that does not capture the physiological results? Why use Euclidean distance if this is clearly not a good measurement? Why not use correlation distance throughout the manuscript? All these questions make the reader lose confidence in the presented results.

4. In Figure 3D, the use of different scales is very confusing and in my opinion misleading. The scale before conditioning is "1", that is, there are basically no connections in the circuit, but this does not look like that in the figure. After conditioning, the scale is "30". From the Figure, due to the different scales, it seems like that the network connectivity becomes sparser, but this is not the case. Actually, the connectivity level is basically nonexistent before training. This also raises the possibility that the observed effects are just because inhibition was added to the circuit. Furthermore, the authors claim that it is reshaping of the network due to conditioning that underlies the decorrelation. However, they did not show what happens with just increased level of inhibition. In fact, in Figure 3 LN-LN and LN-PN connections seem to change exactly in the same manner and in the same cells. This, at least to me, supports the conclusion that the observed changes are not because of reshaping of the network but rather just because of the overall increased inhibition.

5. The authors show bars of mean and error bars. Only in one figure I could find what these error bars are (as far as I could see this was also not stated in the methods). More importantly, it is well known that the average itself is not enough to evaluate data. The authors should show in all figures the actual variance of the results.

6. The results of Figure 6 are only very briefly and trivially discussed. For example, in Figure 6B when odors from Env 1 and Env 2 overlap. Training in Env 1 leads to decorrelation of odors P and Q but not of M and N which belong to Env 2. However, training in Env 2 leads to decorrelation of both P and Q and M and N. So, there is a non-symmetrical relation here. This is completely ignored.

Minor comments:

1. The model only assumes LN-LN and LN-PN connections. While I realize this is the dogma in the field, this is not the case in other insect systems. In particular, this is not the case for the *Drosophila* olfactory system where the vast majority of LN synapses are onto ORNs. I think it is important to explain these differences.

2. The introduction should provide a better literature review of the many findings of inhibitory synaptic plasticity in the insect AL. The authors completely ignore findings from other model organisms that are better studied. More importantly, they ignore already published results regarding the Honeybee. For example, MaBouDi et al. 2017.

3. There are no references to Figures 2C-F.

Reviewer #2

(Remarks to the Author)

Joshi et al. combine neuronal modeling, calcium imaging, and machine learning to investigate the neural mechanisms determining the increase in perceptual distance between rewarded and habituated odorants after a differential conditioning paradigm.

Using protocols of absolute and differential conditioning for training their model, they investigated how the activity of neurons responding to odorants that are either rewarded or subject to habituation evolves with learning. Combining pairs of simulated odorants with different degrees of overlap, they showed that only the neural units responding specifically to the rewarded stimulus increased the firing rate, thus favoring a contrast enhancement strategy for detecting rewarded odorants. Calcium imaging data support these findings, which were also reached using a graph convolutional network for odorant classification.

This work is appropriately designed, well-presented, and clearly written. However, some clarifications would be useful to help the reader better understand the implications of the study.

Minor comments:

line 28: calcium instead of Calcium

line 58: I suggest adding a recent review on the functional and anatomical similarities between vertebrates and invertebrates olfactory lobe (Fulton et al, 2024, Nat Rev Neurosci)

Lines 77-91: I have difficulty in understanding what calcium imaging experiments were done for this study and what was re-used from the Locatelli et al. 2016. The authors should be more straightforward about this. Are they re-analyzing the data or producing a new dataset?

lines 86-91: It is unclear if the difference in ED shown in figure 1D is supported by statistics. If so, test and p-values should be mentioned in this paragraph (and maybe in the figure legend). If, as they say, the change in ED is not significant, the authors should refrain from saying that the ED increases after differential conditioning and adjust their claims here and in the figure caption.

line 134: I am not convinced that the learning - as described here - stretches the AL coding space. I would rather say it places stimuli further apart in the coding space.

line 405: *Apis mellifera* must be in italic

Figure 1C: Add error bars to the plot and n values in the legend (at least a range of n if the values are not the same for each glomerulus). If, instead, 1C shows the glomerular responses of the single individual shown in 1A. In that case, I do not understand why the strongest glomeruli (e.g., in 1C-left) are 47-60-45, which (according to 1B) are not in the position where the most active glomeruli are shown in 1A-left (red-coloured area).

Figure 2B: It is unclear what the seven groups are (both visually and conceptually). The author should clarify this point in the legend but also in the main text for the reader.

Figure 3C: If I understand correctly, the light blue bar indicates the ED between two stimuli in a naive model. As such, the stimuli have not yet been coupled to any habituation or reinforcement. If so, shouldn't the light blue bars be the same (or very similar) when comparing left and right plots? I believe I am missing something here. The authors should explain better this plot - which is very relevant for results and discussion - to avoid misunderstandings.

Figure 3D: the second plot color bar overlaps with the y-axis label of the third plot.

As a general comment, the author should be uniform in their writing style. Some examples are the usage of honey bee or honeybee throughout the manuscript (e.g., lines 42 and 47); the use of antennal lobe and olfactory bulb should always be in lower case instead of mixing upper and lower case (see lines 54 - 55); units should be either always attached or detached from the number (e.g., 35ms or 35 ms); calcium imaging or Calcium Imaging?

Figure 3B: On the same line as the previous comment, "odor" and "trajectories" are simple names. As such, they should all start with a capital letter or not. Uniform the legend, please. Also, in the figure caption, "habituated odors" is written as "Habituated (or Rewarded) Odors" in some instances. I suggest sticking to lowercase for common names.

Version 1:

Reviewer comments:

Reviewer #1

(Remarks to the Author)

The authors have adequately answer most of my concerns. The manuscript is now greatly improved.

I still think the authors should tone down their claim that their model is fully realistic. The results and conclusions are not diminished by saying that the model does not fully recapitulate the physiological properties of the circuit. Nevertheless, I leave this to the authors and editor to decide.

Reviewer 1

Reviewer #1 (Remarks to the Author): In this manuscript Joshi et al. use computational modelling to try and demonstrate that associative and non-associative plasticity in the honeybee antennal lobe (AL) work together to enhance the contrast between rewarded and unrewarded odors. In general, this manuscript has the potential to contribute to the field. However, there are some major concerns that reduce my enthusiasm for the current version of the manuscript.

We would like to thank the reviewer for their questions and suggestions, which greatly helped to improve our work.

Major comments:

1. Comment 1:

The authors attempt to provide physiological evidence to the construction and execution of the model. However, in some cases these justifications are inaccurate, and, in some cases, physiological evidence is ignored. For example, the authors present “odor” responses of PNs and LNs (Figure 2C, D). *However, these responses are not similar at all to published recordings from the honeybee (See for example Meyer et al. 2013, Krofczik et al. 2009).* From the presented figures, PNs and LNs have very low firing rates but in reality, at least PNs can reach very high firing frequencies. Similarly, PNs and LNs are modeled to have a rather step-like response, whereas, in reality, the response is highly dynamic. Another point related to the model construction is the ORN input and its transformation to PNs and LNs. PNs and LNs are simulated in the same manner and with the same response profile. However, at least in flies, this was shown not to be the case. *Rather, PNs have a non-linear response to ORN input whereas LNs have a linear response.* Furthermore, LNs hardly respond to low odor concentration whereas PNs do. I am not implying that every biological feature needs to be modeled, but rather that the model should have been more carefully and accurately constructed. In honeybees there are five octopamine receptors. The authors cite a manuscript that showed that one type of octopamine receptor is expressed in GABAergic LNs and therefore they conclude that octopamine exerts its effect in presynaptic terminals of LNs. I find this a bit of a stretch. Similarly, in flies Oct β 1r and Oct β 2R were reported to have opposite effects. Furthermore, it was suggested that Oct β 1r reduces cAMP levels whereas Oct β 2R increases cAMP levels (Koon and Budnik, JNS 2012). Finally, cAMP level by itself is not an indication to whether excitation or inhibition eventually occurs. How can one conclude that octopamine facilitates LN presynaptic activity? Also, *Amoa1* was reported to increase overall glomerular activity (Rein et al. 2013). The paper (Rein et al.) that is used to support the assumption that OA modulates LN

presynaptic terminals actually states: “Rather, it is likely driven by external factors, e.g. by decreased inhibitory input” – thus they reach the opposite conclusion about OA modulation of LN activity. To summarize this part, this is an interesting model, but it cannot be presented as a realistic model.

Response to comment 1:

We thank the reviewer for their thoughtful and detailed feedback which helped to improve our manuscript. While our model necessarily simplifies some biological details, we believe it captures the essential features of the antennal lobe network dynamics required to study learning-induced changes in odor representations.

Specifically, regarding the model construction and physiological responses, we rarely see honeybee PN firing rates exceeding 40-60 Hz in vivo, at least for low-to-middle range of odor concentrations. In Figures 1 and 2 of this response letter (now included as new Suppl Figures), we present analysis of PN responses in vivo (data from Smith lab), as well as the model PN responses. While firing rates of the model neurons are slightly lower than in vivo, both are largely distributed 0-20 Hz for background activity (Fig. 2, top) and 0-30(40) Hz during model (in vivo) odor response (Fig. 2, middle). It is also worth mentioning that in vivo recordings are naturally biased towards more active neurons, as those are more likely to be selected by spike sorting algorithms. And in vivo neurons are not grouped by response types, so the neurons which reduce their activity during odor stimulation are present but harder to see in the raster plots. Finally, we would like to note that the key phenomena we are investigating - learning-induced changes in odor representations - primarily depend on neural activity during odor presentation when neuromodulatory signals associated with reward are active. The background activity before odor onset plays a relatively minor role in this specific context.

While in vivo responses are undoubtedly more complex and dynamic than those generated by model, our experimental recordings show that PN responses to odor stimulation often display step-like characteristics similar to those produced by our model (Figure 1 of this letter). Still, our model responses are simplified, and this is related to important limitation of our model - we do not have direct access to the receptor neuron response patterns and must therefore work with simplified approximations. We mentioned these important limitations in the revised text on line 135 in the results section:

“While our model dynamics reasonably match in vivo data (see Supplementary Fig. 1), PN firing rates in the model are slightly lower than those observed in vivo, particularly during odor stimulation and post-odor response. It is important to note, however, that in vivo recordings are naturally biased towards

more active neurons. Additionally, model PN responses exhibit somewhat simplified temporal patterns, which is related to the lack of access to the simultaneous ORN recordings. Therefore, we used simplified approximations of the odor input, similar to our previous work in locusts (Bazhenov et al. 2001), moths (Ito et al. 2009), and honeybees (Chen et al. 2015)."

Regarding the concern about LN responses to different odor concentrations, while this is an important aspect of AL function, our current study focuses on learning-induced changes at fixed concentrations. Therefore, the concentration-dependent aspects of LN responses, while interesting and important, fall outside the scope of our current investigation.

Finally, we would like to thank the reviewer for bringing up a very important question about octopamine receptors. The reviewer is correct that there are several octopamine receptors in honeybees and fruit flies. The two mentioned - Oct β 1r and Oct β 2R – induce changes in cAMP, and it is clear that changes in cAMP alone cannot predict the excitatory/inhibitory effect of octopamine binding to those receptors. It is as yet unknown where these receptors are expressed in the honey bee CNS. However, the receptor that we (Smith lab; Sinakevitch et al. 2011) have worked with is AmOA1, and its ortholog in *Drosophila* is DmOAMB (note that there are different abbreviations for these receptors – e.g. DmOA1). For this receptor, one consequence in honey bees (Grohmann et al. 2003) and fruit flies (Lee et al. 2003) of octopamine binding to it is an oscillatory increase in intracellular Ca²⁺, which will presumably be excitatory. (Sinakevitch, Mustard, and Smith 2011) also report that the receptor is present *on a subset* (i.e., not on all) of the inhibitory interneurons in the antennal lobe of honeybees and fruit flies. The precise interconnectedness of inhibitory local interneurons in the antennal lobe is not known yet. But it is conceivable that the subset excited by octopamine via AmOA1 inhibits other inhibitory interneurons that connect to PNs, which would produce the PN disinhibition reported by Smith and colleagues in (Rein et al. 2013). Excitation via inhibition of inhibition occurs in the CNS of many animals. We have not proven it here, but our model now proposes testable hypotheses based on it. We included a brief discussion of this important topic in Results on line 112:

"Here we focus on AmOA1 octopamine receptor (Sinakevitch et al., 2011) expressed alongside GABA receptors in the honeybee AL (Sinakevitch et al., 2013). In honeybees (Grohmann et al., 2003) and fruit flies (Lee et al., 2003), octopamine binding to this receptor leads to an increase in intracellular Ca²⁺ that primes adenylyl cyclase to enhance cAMP and cAMP-dependent protein kinase activation that further regulates synaptic plasticity" and in Discussion on line 393:

“While different types of octopamine receptors, such as OctB1R and OctB2R, are expressed in the honeybee brain, here we focus on AmOA1 (Sinakevitch, Mustard, and Smith 2011), which is orthologous to *Drosophila*’s DmOAMB receptor. In honeybees (Grohmann et al. 2003) and fruit flies (Lee et al. 2003), octopamine binding to this receptor leads to an increase in intracellular Ca^{2+} , presumably resulting in an excitatory effect. Importantly, this receptor is present on a subset (i.e., not all) of the inhibitory interneurons in the antennal lobe of honeybees and fruit flies (Sinakevitch, Mustard, and Smith 2011). Thus, activation of these receptors on a subset of LNs may inhibit other LNs and thereby lead to partial PN disinhibition, as reported in (Rein et al. 2013).”

Figure 1: Responses of PNs before (left) and after (right) conditioning, in both the in vivo data (top) and model data (bottom). The odor was presented for 500ms starting from 0.5s to 1s in the raster plot.

Figure 2: Histograms of firing rates for in vivo and model PN responses for before and after conditioning. The three rows show firing rates calculated during different time periods. Row 1 shows firing rates for background, spontaneous activity. Row 2 shows firing rates calculated during 500ms of odor stimulation. Row 3 shows firing rates calculated 1s after the offset of the odor stimulation.

2. Comment 2:

The authors do not show an exploration of the model parameter space. The model's parameters seem to be very specific. How they were selected was not explained. More importantly, how sensitive are the conclusions to changes in the model's parameters was not demonstrated. Do slight changes in the parameters completely change the model outcomes? This could perhaps be negated in a very realistic model in which its parameters are completely based on measured results. However, as explained in point 1 above, this is not the case here.

Response to comment 2:

We thank the reviewer for this important point about parameter space exploration. We have now conducted a comprehensive sensitivity analysis of key model parameters to evaluate the robustness of our findings. Specifically, we systematically varied all the following critical parameters:

1. GABAergic synaptic conductances:
 - $g_{\text{GABA_A}}$ (fast GABA inhibition)
 - $g_{\text{GABA_B}}$ (slow GABA inhibition)
2. Excitatory synapses:
 - g_{ACh} (fast ACh excitation)
3. Input synaptic conductances:

- Extern_ach1 (input conductance for PNs)
 - Extern_ach2 (input conductance for LNs)
4. Plasticity parameters:
 1. dF (facilitation rate)
 2. TrF (weight decay time constant)

For each parameter, we performed perturbations around the baseline values used in our original simulations while keeping other parameters fixed and conducted differential conditioning and testing using the method described in the manuscript. We calculated the total number of spikes before and after conditioning as well as the correlation values between odor classes A and B. In Fig 3 below (now included as new Suppl Figure), we show effects of perturbations for cases of fast GABA and slow GABA conductances, for the facilitation rate (dF), and for the input conductance for PNs (Extern_ampa1). Our analysis for all the parameter perturbations revealed that the model's core behavior - enhanced separation between rewarded and habituated odor representations after learning, as shown by a decrease in correlation similarity between the odor classes after differential conditioning - remains qualitatively consistent across a broad range of parameter values. Small variations in these parameters produced only minor quantitative changes in neuronal firing rates and did not alter the fundamental decorrelation effect.

We have modified the main text to address this point (see specific changes below). We have included a new supplementary figure showing the changes in spike counts across parameter perturbations and the correlation between representations of odors A and B before and after conditioning for different parameter values. The following text was included in Results on line 155:

“To evaluate the robustness of our findings, we have conducted a sensitivity analysis of the key model parameters. Specifically, we systematically varied the following: fast and slow GABAergic and ACh synaptic conductances between LNs and PNs, input synaptic conductances to LNs and PNs, synaptic plasticity parameters (facilitation rate and decay time constant). We found that the model's core behavior - enhanced separation between rewarded and habituated odor representations after learning - remains qualitatively consistent across a broad range of parameter values (Suppl Figure2).”

Figure 3: Parameter sensitivity analysis demonstrates robustness of model behavior. (A-D) Each panel shows the effect of perturbing a different parameter: (A) fast GABA inhibition (g_{GABA_A} , base value = $2.0e-5$), (B) slow GABA inhibition (g_{GABA_B} , base value = $2.0e-4$), (C) facilitation rate (dF , base value = $1.5e-1$), and (D) input synaptic conductance (g_{Extern_ach1} , base value = $9.0e-6$). For each parameter, the top row shows the average spike count across all PNs before (left) and after (right) differential conditioning, while the bottom row shows the correlation between odor classes A and B before (left) and after (right) differential conditioning. The x-axis represents the percent change from the baseline parameter value. Despite variations in absolute spike counts, the key finding of reduced correlation between odor representations after differential conditioning remains robust across parameter perturbations, indicating that the model's core behavior does not depend on precise parameter tuning.

3. Comment 3:

The authors use a confusing approach regarding the matrices they are using. In Figure 1 they show the non-significant Euclidean distance result and not the significant correlation result that the authors mention they have. Later, they use the model to show significant Euclidean distance results. So, what are we to believe in? The physiological results that show no significant difference between absolute and conditional conditioning, the model results that do show a significant effect or the results that were not even presented in the manuscript. What can we learn from a model that does not capture the physiological results? Why use Euclidean distance if this is clearly not a good measurement? Why not use correlation distance throughout the manuscript? All these questions make the reader lose confidence in the presented results.

Response to comment 3:

We thank the reviewer for this important observation. We agree that our initial approach of using Euclidean distance metrics created unnecessary confusion. In our revision, we have addressed this by consistently using correlation-based similarity measures throughout the manuscript, which aligns with the significant physiological results reported in (Locatelli, Fernandez, and Smith 2016). We have made the following changes:

1. Removed the non-significant Euclidean distance results from Figure 1 and replaced it with correlation similarity (see fig 4 below, which is new fig 1 in the manuscript). This correlation similarity was calculated similar to what was reported in (Locatelli, Fernandez, and Smith 2016).
2. Replaced all Euclidean distance measurements with correlation-based similarity metrics throughout the manuscript. Note, all the model results previously reported as significant using Euclidian distance remain significant using correlation similarity.
3. Ensured consistency between our model predictions and the physiological data, which shows significant changes in correlation-based similarity after differential conditioning.

Below is the updated version of the Figure 1 in the manuscript with correlation similarity instead of Euclidean distance.

Figure 4 (Updated Figure 1 in manuscript): Olfactory representations in the honeybee AL. A) Calcium imaging responses elicited by two different odors (PH1 and PP1) in a representative bee. Here, PH1 and PP1 belong to different classes of odors, with PH odors being presented along with a reward (rewarded odors) and PP odors presented without any reward (habituated odors). B) Position of the glomeruli within the Antennal Lobe, for which Calcium Imaging responses were measured. C) Mean activation across the different glomeruli elicited by the PH1 and PP1 odors. Glomeruli are ordered with the glomeruli most activated by PH1 at the top and the glomeruli least activated by PH1 at the bottom. The same ordering is used for PP1 odor as well to highlight the difference in the activation patterns between the two odors. D) Correlation similarity of glomeruli activation between different odor classes (PH and PP) decreases after differential conditioning compared to absolute conditioning. This indicates that the glomerular representation gets more separable after differential conditioning (Wilcoxon signed rank test, $p=4.8e-8$, the error bars show SEM). E) Honeybee training protocol for differential and absolute conditioning. In differential conditioning, the bee was trained with rewarded (PH) and habituated (PP) odors presented in a randomized order. For absolute conditioning, the bee was trained with only rewarded odors. Mineral Oil (MO) was presented to the bee without any reward during absolute conditioning. Each trial lasts for 4s, with 500ms of odor presentation (Data from Locatelli et al (2016)). The same training protocol was used for training the computational model.

4. Comment 4:

In Figure 3D, the use of different scales is very confusing and in my opinion misleading. The scale before conditioning is “1”, that is, there are basically no connections in the circuit, but this does not look like that in the figure. After conditioning, the scale is “30”. From the Figure, due to the different scales, it seems like that the network connectivity becomes sparser, but this is not the case. Actually, the connectivity level is basically nonexistent before training. This also raises the possibility that the observed effects are just because inhibition was added to the circuit. Furthermore, the authors claim that it is reshaping of the network due to conditioning that underlies the decorrelation. However, they did not show what happens with just increased level of inhibition. In fact, in Figure 3 LN-LN and LN-PN connections seem to change exactly in the same manner and in the same cells. This, at least to me, supports the conclusion that the observed changes are not because of reshaping of the network but rather just because of the overall increased inhibition.

Response to comment 4:

We thank the reviewer for this comment. The total amount of inhibition in the network does indeed increase after training. To address the question of whether the decorrelation effects arise from specific network restructuring versus general increase of inhibition, we conducted a control experiment:

We performed network shuffling analyses where we preserved the total inhibitory strength after training while randomizing the specific connectivity patterns in both LN-LN and LN-PN networks (see figure 5 at the end of this comment, which is now included as new Suppl Figure). Specifically, we maintained identical distribution of the weights in the shuffled networks but disrupted the structured connectivity patterns that emerged during learning. These shuffled networks, despite having equivalent total inhibitory drive, failed to produce the between and within class odor decorrelation that was observed in the structured network. This demonstrates that the specific spatial organization of inhibitory connectivity, not merely the overall inhibition level, is essential for the observed learning effects. The following text was included in Results on line 191:

“The training increased the total amount of inhibition in the network. To verify that the odor-specific structure of inhibitory connections, rather than the overall increase in inhibition, aids odor discrimination, we conducted a shuffling experiment. Specifically, we maintained an identical distribution of the weights in the shuffled networks but disrupted the structured connectivity patterns that emerged during learning. These shuffled networks, despite having an equivalent total inhibitory drive, failed to produce the between- and within-class odor decorrelation observed in the structured network (see Supplementary Figure 3). This supports our prediction that enhanced odor discrimination emerges from the precise restructuring of inhibitory circuits during learning, rather than from global increases in inhibition.”

Figure 5: Analysis of network connectivity structure and odor discrimination. A) Correlation between odor representations comparing naive, associative-only, non-associative-only, and combined (differential) conditioning for odors between different classes (left) and for odors within the same class (right). B) Connectivity matrices showing shuffled LN-LN and LN-PN networks after differential conditioning. When the structured connectivity patterns are disrupted through shuffling while maintaining the same total inhibitory strength, the enhanced odor discrimination is lost, demonstrating that the specific spatial organization of inhibitory connections, not just increased inhibition, is essential for improved odor discrimination. Side bars indicate percept identities (green: unique to rewarded odor, orange: shared, red: unique to habituated odor). Color bars show relative synaptic weights.

5. Comment 5:

The authors show bars of mean and error bars. Only in one figure I could find what these error bars are (as far as I could see this was also not stated in the methods). More importantly, it is well known that the average itself is not enough to evaluate data. The authors should show in all figures the actual variance of the results.

Response to comment 5:

Thank you for this comment. While the statistical tests and standard deviations were included in the main text, their presentation could be more accessible. We have now added comprehensive statistical information to all figure captions, including:

- Standard deviation for all bar plots
- Specific statistical tests used
- Exact p-values for all comparisons

6. Comment 6:

The results of Figure 6 are only very briefly and trivially discussed. For example, in Figure 6B when odors from Env 1 and Env 2 overlap. Training in Env 1 leads to decorrelation of odors P and Q but not of M and N which belong to Env 2. However, training in Env 2 leads to decorrelation of both P and Q and M and N. So, there is a non-symmetrical relation here. This is completely ignored.

Response to comment 6:

We thank the reviewer for bringing attention to the brevity of our discussion regarding the environmental learning results. Following this feedback, we have substantially expanded our analysis and presentation of these findings. We have updated Figure 6 to use correlation similarity instead of Euclidean distance, as this metric better captures the relationships between odor representations, and we have revised the accompanying text. Our main results do not change when we use correlation similarity vs Euclidean distance here. To answer the reviewer's question regarding Figure 6B, a non-symmetric relationship between environments emerges from a few factors.

First, we would like to emphasize that the model includes forgetting mechanism where synaptic weights gradually return to their original values when there are no synaptic updates (see Methods; we now emphasized this explicitly in the Results of the revised version). Thus, e.g., in Fig. 6A, when the network is initially trained on Env1 (P+,Q-) leading to decorrelation between P and Q odor representations, the correlation between P and Q starts to increase during training on the 2d non-overlapping Env2 (S+,T-). Second, forgetting mechanisms is slow and the network does not fully recover to its original naïve state between trainings on Env1 and Env2. Thus, in scenario of the Fig. 6B, training naïve network on Env1 affects correlation between odors M and Q differently than training Env2 affects correlation between P and Q. In the last case, the effect depends on several specific factors, such as which odor components overlap and how network excitability was affected by initial training on Env 1.

The following is the updated text and figure in the results section:

On line 258, we have added:

“Finally, we included a mechanism where synaptic weights return to baseline in the absence of reinforcement. This “forgetting” mechanism was slow so the network

does not fully recover to its original naïve state between trainings on Env1 and Env2. Because of that, in some cases, effects of training Env 1 and Env 2 were asymmetric as Env 2 was applied to the network that was already pretrained on Env1”.

Starting from line 263, we have edited the text:

“ *Case 1: When there is no overlap between odors from Env1 and Env2, the correlation between representations of classes P and Q decreases significantly after training on Env1 (P+Q-) but increases again after training on Env2 since the weights associated with discriminating between P and Q odors receive no reinforcement and slowly return to baseline. The correlation between odors S and T only decreases after training on Env2, regardless of reward structure (S+T- or S-T+), indicating effective learning in the new environment without interference from prior learning (Figure 6A).*

Case 2: When both rewarded and habituated odors overlap between environments (Figure 6B), we observe that if the reward structure remains consistent (M+N-) (Figure 6B, left), the correlation between P and Q representations remains low after training on Env2, reinforcing the learning from Env1. This happens because of the existing memory trace of Env1 and the overlap maintains a fraction of the weights supporting discrimination between odors P and Q. However, when reward associations are reversed (M-N+) (Figure 6B, right), the decorrelation between P and Q increases again to the naïve level, more than in the case of non-overlapping odors, suggesting interference between the two environments.

Case 3: When only the rewarded odors overlap (class M overlaps with class P), the correlation between P and Q remains low when reward structure of Env2 is consistent with Env1 (Fig. 6C, left,) but significantly increases when reward structure of Env2 is opposite to Env1 (Fig. 6C, right). The network is able to learn new associations between M and S regardless of the reward structure of Env2 (Figure 6C).

Case 4: When odors from Env2 (class N) only overlap with the habituated odors from Env1 (class Q), we observe an increase in correlation between P and Q after training on Env2 regardless of its rewards structure (Figure 6D). The network is also able to learn new associations between S and N. This suggests that changing the reward association of previously habituated components has minimal effect on existing memories compared to cases where rewarded components change their associations.

These results demonstrate that the AL network can effectively learn new odor-reward associations while maintaining existing ones, particularly when there is minimal overlap between environments or when reward structures remain consistent. The network shows more flexibility in adapting to new reward associations when they don't directly conflict with strongly learned rewarded odor components from the previous environment.”

Figure 4 (Updated Figure 6 in manuscript): Learning by the AL inhibitory network in different environments. The network is initially trained on environment 1 (Env1) containing rewarded odor class P and habituated odor class Q. Subsequently, the network is trained on a new environment 2 (Env2), containing different odor classes. Bars show the correlation between odor representations for classes P and Q (left set of bars in each subplot) and between odor classes from Env2 (right set of bars) at different stages: naive (blue), after training on Env1 (orange), and after subsequent training on Env2 (yellow). Each case (1-4) represents a different scenario of odor overlap between Env1 and Env2, with sub-cases (left and right subplots for each case) depicting different reward structures in Env2. A) Case 1: No odor overlap between environments (Env2 odors: S,T). B) Case 2: Complete overlap - both odors from Env2 (M,N) overlap with odors from Env1 (P,Q respectively). C) Case 3: Partial overlap - one odor from Env2 (M) overlaps with the rewarded odor from Env1 (P), while the second odor (S) has no overlap. D) Case 4: Partial overlap - one odor from Env2 (N) overlaps with the habituated odor from Env1 (Q), while the second odor (S) has no overlap. The boxes at the top illustrate the distinct percepts activated by the odors in each environment, with overlapping percepts indicated by shared columns. Lower correlation values indicate better separation between odor representations. Error bars show standard deviation across multiple trials.

Minor comments

1. The model only assumes LN-LN and LN-PN connections. While I realize this is the dogma in the field, this is not the case in other insect systems. In particular, this is not the case for the *Drosophila* olfactory system where the vast majority of LN synapses are onto ORNs. I think it is important to explain these differences.

Response: Thank you for raising this important point about the network connectivity. Indeed, our model focuses on the canonical AL circuit and includes

three types of intrinsic connections, i.e., LN->PN, LN->LN, PN->LN (see Table 1 below), as described in the honeybee AL. While we acknowledge the importance of the other synaptic connections, our focus on inhibitory plasticity (LN-to-PN and LN-to-LN synapses) reflects our central hypothesis that experience-dependent modifications of inhibitory circuits reshape odor representations. Nevertheless, we agree that LN-mediated presynaptic inhibition of the ORN axons, as found in *Drosophila*, would add another layer of complexity to the AL response dynamics and affect how odor discrimination is shaped by associative learning.

Table 1 Connection probabilities between different neuron groups in the AL.

Source Unit	Target Unit	Connection Probability
All PNs	All PNs	0
	Unipolar LNs	0.4
	Multipolar LNs	0.4
Unipolar LNs	PNs from same glomerulus	0
	PNs from different glomerulus	0.5
	Unipolar LNs from same glomerulus	0
	Unipolar LNs from different glomerulus	0.4
	Multipolar LNs	0.3
Multipolar LNs	All PNs	0.3
	Multipolar LNs	0.1
	Unipolar LNs	0.4

We included the following text to Discussion on line 418 to address this model limitation:

“Our AL model focuses on the synaptic connections between excitatory PNs and inhibitory LNs. While those represent canonical AL circuit, other types of connections are found in different insect species, e.g., in fly olfactory system the vast majority of LN also synapses onto ORNs (Olsen and Wilson 2008). While we acknowledge this model limitation, our focus on inhibitory plasticity (LN-to-PN and LN-to-LN synapses) reflects our central hypothesis that experience-dependent modifications of inhibitory circuits reshape odor representations in the AL.”

2. The introduction should provide a better literature review of the many findings of inhibitory synaptic plasticity in the insect AL. The authors completely ignore findings from other model organisms that are better studied. More importantly, they ignore already published results regarding the Honeybee. For example, MaBouDi et al. 2017.

Response: We appreciate the reviewer's suggestion to provide a more comprehensive review of inhibitory plasticity across insect model systems. We have expanded our introduction to include key findings about inhibitory synaptic plasticity in various insect antennal lobes, particularly drawing from well-studied model organisms like *Drosophila*. We have also strengthened our coverage of previous experimental findings in the honeybee AL, including work demonstrating

experience-dependent changes in inhibitory networks (MaBouDi et al. 2017). These additions provide important context for understanding how our computational model builds on and extends previous experimental observations across species. The following text was added in the Introduction on line 62:

“Studies across different insect species have revealed conserved mechanisms of plasticity in early olfactory processing. In Drosophila, prolonged odor exposure leads to selective potentiation of inhibitory synapses between local interneurons and projection neurons, resulting in habituation (Das et al. 2011). In moths, associative conditioning modifies both excitatory and inhibitory connections within the antennal lobe network, enhancing responses to behaviorally relevant odors (Daly, Wright, and Smith 2004). Work in locusts has demonstrated that plasticity in the antennal lobe can shape representations to improve stimulus discrimination (Bazhenov et al. 2005). In honeybees, both associative and non-associative forms of plasticity have been shown to modify AL network dynamics, with computational models suggesting that such modifications can enhance discrimination between similar odors (MaBouDi et al. 2017; Marachlian, Klappenbach, and Locatelli 2021; Locatelli et al. 2013; Locatelli, Fernandez, and Smith 2016). Together with findings from other insects, these studies highlight inhibitory network plasticity as a conserved mechanism for optimizing early olfactory processing across insect species ”

3. There are no references to Figures 2C-F.

Response: We thank the reviewer for noting this oversight. We have added explicit references to Figures 2C-F in the Results section of the manuscript. These figures illustrate our model's basic properties: Figure 2C shows the spiking activity of a representative projection neuron (PN), Figure 2D demonstrates the activity pattern of a local interneuron (LN), Figure 2E displays the local field potential (LFP) averaged across the PN population, and Figure 2F shows the temporal structure of the input current pulse provided to the neurons during odor stimulation. The revised text on the results section on line 122 now says:

“The spiking activity of a representative PN and LN are shown in Figure 2C, D. Figure 2E shows the local field potential (LFP) obtained by averaging membrane voltages across the PN population and Figure 2F shows temporal structure of the input current pulse to the neurons during odor stimulation. The amplitude of this current pulse to each neuron depends on the odor currently being presented. ”

Reviewer 2

Reviewer #2 (Remarks to the Author): Joshi et al. combine neuronal modeling, calcium imaging, and machine learning to investigate the neural mechanisms determining the

increase in perceptual distance between rewarded and habituated odorants after a differential conditioning paradigm. Using protocols of absolute and differential conditioning for training their model, they investigated how the activity of neurons responding to odorants that are either rewarded or subject to habituation evolves with learning. Combining pairs of simulated odorants with different degrees of overlap, they showed that only the neural units responding specifically to the rewarded stimulus increased the firing rate, thus favoring a contrast enhancement strategy for detecting rewarded odorants. Calcium imaging data support these findings, which were also reached using a graph convolutional network for odorant classification. This work is appropriately designed, well-presented, and clearly written. However, some clarifications would be useful to help the reader better understand the implications of the study.

We would like to thank the reviewer for their interest in our study and for highlighting the importance of our work.

Minor Comments:

1. **line 28:** calcium instead of Calcium

Response: Thank you for catching this inconsistency. We have corrected "Calcium" to "calcium" and standardized the capitalization throughout the manuscript.

2. **line 58:** I suggest adding a recent review on the functional and anatomical similarities between vertebrates and invertebrates olfactory lobe (Fulton et al, 2024, Nat Rev Neurosci)

Response: We appreciate the suggestion to include this recent and highly relevant review. We have added the citation to (Fulton et al. 2024) which strengthens our discussion of the functional and anatomical similarities between vertebrate and invertebrate olfactory systems. The text in the introduction on line 58 now says: *“Together with the now well-established similarities in anatomical connectivity within the AL and OB networks (Fulton et al. 2024, ...) ...”*

3. **Lines 77-91:** I have difficulty in understanding what calcium imaging experiments were done for this study and what was re-used from the Locatelli et al. 2016. The authors should be more straightforward about this. Are they re-analyzing the data or producing a new dataset?

Response: We thank the reviewer for pointing out this ambiguity. We are re-analyzing the calcium imaging dataset from (Locatelli, Fernandez, and Smith 2016) rather than presenting new experimental data. This reanalysis provides new insights into the contrast enhancement mechanism we describe. We have updated the text to state this more clearly. The new text in the results section on line 222 says:

“The calcium imaging data from honeybees has been published previously in (Locatelli, Fernandez, and Smith 2016) and here we re-analyze this data.”

4. **Lines 86-91:** It is unclear if the difference in ED shown in figure 1D is supported by statistics. If so, test and p-values should be mentioned in this paragraph (and maybe in the figure legend). If, as they say, the change in ED is not significant, the authors should refrain from saying that the ED increases after differential conditioning and adjust their claims here and in the figure caption.

Response: Thank you for this important point. We have revised our analysis to use correlation similarity rather than Euclidean distance throughout the manuscript. We have updated Figure 1 to show the correlation analysis and added the corresponding statistical tests and p-values to the figure caption (see below).

Figure 1 (Updated Figure 1 in manuscript): Olfactory representations in the honeybee AL. A) Calcium imaging responses elicited by two different odors (PH1 and PP1) in a representative bee. Here, PH1 and PP1 belong to different classes of odors, with PH odors being presented along with a reward (rewarded odors) and PP odors presented without any reward (habituated odors). B) Position of the glomeruli within the Antennal Lobe, for which Calcium Imaging responses were measured. C) Mean activation across the different glomeruli elicited by the PH1 and PP1 odors. Glomeruli are ordered with the glomeruli most activated by PH1 at the top and the glomeruli least activated by PH1 at the bottom. The same ordering is used for PP1 odor as well to highlight the difference in the activation patterns between the two odors. D) Correlation similarity of glomeruli activation between different odor classes (PH and PP) decreases after differential conditioning compared to absolute conditioning. This indicates that the glomerular representation gets more separable

after differential conditioning (Wilcoxon signed rank test, $p=4.8e-8$, the error bars show SEM). E) Honeybee training protocol for differential and absolute conditioning. In differential conditioning, the bee was trained with rewarded (PH) and habituated (PP) odors presented in a randomized order. For absolute conditioning, the bee was trained with only rewarded odors. Mineral Oil (MO) was presented to the bee without any reward during absolute conditioning. Each trial lasts for 4s, with 500ms of odor presentation (Data from Locatelli et al (2016)). The same training protocol was used for training the computational model.

5. **line 134:** I am not convinced that the learning - as described here - stretches the AL coding space. I would rather say it places stimuli further apart in the coding space.

Response: We replaced "stretches the coding space" with "places stimuli further apart in the coding space" throughout the manuscript.

6. **line 405:** *Apis mellifera* must be in italic

Response: We have italicized *Apis mellifera* throughout the manuscript following standard scientific notation.

7. **Figure 1C:** Add error bars to the plot and n values in the legend (at least a range of n if the values are not the same for each glomerulus). If, instead, 1C shows the glomerular responses of the single individual shown in 1A. In that case, I do not understand why the strongest glomeruli (e.g., in 1C-left) are 47-60-45, which (according to 1B) are not in the position where the most active glomeruli are shown in 1A-left (red-colored area).

Response: Figure 1C shows glomerular activity patterns from a single representative bee rather than population averages. We have clarified this in the revised text. We have changed Figure 1B since the diagram did not accurately correspond to the calcium imaging data shown in Figure 1A.

8. **Figure 2B:** It is unclear what the seven groups are (both visually and conceptually). The author should clarify this point in the legend but also in the main text for the reader.

Response: The odors in the current work comprise of different 'pure' chemicals and each 'pure' chemical activates one group of PNs out of the 7. For the simple odors shown in Figure 2B, each odor activated 3 groups of PNs out of 7, with different amount of overlap between them. We have modified the text in the methods section on line 566 to better explain the 7 groups:

"Each odor was modelled as a mixture of 'pure' chemicals. Each pure chemical is defined here as a percept. Each percept activates a group of 'virtual ORNs', which in turn activate a set of PNs. The odors we created activate 3 out of 7 percepts, which means each odor is a mixture of 3 'pure' chemicals out of a total of 7 'pure' chemicals."

We also modified the legend to Figure 2B to read:

“B) Spatial pattern of odor stimuli for rewarded and habituated odors. The PNs are split into 7 groups, each representing one odor percept: Percept 1 = [PN1 - PN14]; Percept 2 = [PN15 - PN28]; ... Each simulated odor activates three selected percepts. In this experiment, percepts 1,2 and 3 are activated by the rewarded odor (green), while percepts 2,3 and 4 are activated by the habituated odor (red). The rectangles on top of the X-axis show the glomerulus number, which are used to set up anatomical connectivity between LNs and PNs (see Methods).”

9. **Figure 3C:** If I understand correctly, the light blue bar indicates the ED between two stimuli in a naive model. As such, the stimuli have not yet been coupled to any habituation or reinforcement. If so, shouldn't the light blue bars be the same (or very similar) when comparing left and right plots? I believe I am missing something here. The authors should explain better this plot - which is very relevant for results and discussion - to avoid misunderstandings.

Response: Thank you for raising this point about Figure 3C. We have updated all analyses to use correlation similarity instead of Euclidean distance. The difference in the light blue bars between left and right plots reflects that odors from the same class are inherently more similar to each other than odors from different classes, even in the naive condition. We have updated the results text on line 164 as follows: *“We found a significant decrease in correlation after differential training when comparing rewarded vs. habituated odors and a modest decrease when comparing odors within the same class, i.e., just rewarded or just habituated odors (Figure 3C). Note that odors from the same class are inherently more similar to each other than odors from different classes, even in the naive condition.”*

10. **Figure 3D:** the second plot color bar overlaps with the y-axis label of the third plot.

Response: We have fixed the overlapping colorbar and axis label as suggested.

11. As a general comment, the author should be uniform in their writing style. Some examples are the usage of **honey bee or honeybee** throughout the manuscript (e.g., lines 42 and 47); the use of antennal lobe and olfactory bulb should always be in lower case instead of mixing upper and lower case (see lines 54 - 55); units should be either always attached or detached from the number (e.g., 35ms or 35 ms); calcium imaging or Calcium Imaging? **Figure 3B:** On the same line as the previous comment, "odor" and "trajectories" are simple names. As such, they should all start with a capital letter or not. Uniform the legend, please. Also, in the figure caption, "habituated odors" is written as "Habituated (or Rewarded) Odors" in some instances. I suggest sticking to lowercase for common names.

Response: We greatly appreciate these detailed suggestions. We have standardized our usage throughout the manuscript for:

- "honeybee" as one word
- "antennal lobe" and "olfactory bulb" in lowercase
- Consistent formatting of units
- "calcium imaging" in lowercase
- Consistent capitalization in figure legends

References

- Bazhenov, Maxim, Mark Stopfer, Mikhail Rabinovich, Ramon Huerta, Henry D.I. Abarbanel, Terrence J. Sejnowski, and Gilles Laurent. 2001. "Model of Transient Oscillatory Synchronization in the Locust Antennal Lobe." *Neuron* 30 (2): 553–67. [https://doi.org/10.1016/S0896-6273\(01\)00284-7](https://doi.org/10.1016/S0896-6273(01)00284-7).
- Bazhenov, Maxim, Mark Stopfer, Terrence J. Sejnowski, and Gilles Laurent. 2005. "Fast Odor Learning Improves Reliability of Odor Responses in the Locust Antennal Lobe." *Neuron* 46 (3): 483–92. <https://doi.org/10.1016/j.neuron.2005.03.022>.
- Chen, Jen-Yung, Emiliano Marachlian, Collins Assisi, Ramon Huerta, Brian H. Smith, Fernando Locatelli, and Maxim Bazhenov. 2015. "Learning Modifies Odor Mixture Processing to Improve Detection of Relevant Components." *The Journal of Neuroscience* 35 (1): 179–97. <https://doi.org/10.1523/JNEUROSCI.2345-14.2015>.
- Daly, Kevin C., Geraldine A. Wright, and Brian H. Smith. 2004. "Molecular Features of Odorants Systematically Influence Slow Temporal Responses Across Clusters of Coordinated Antennal Lobe Units in the Moth *Manduca Sexta*." *Journal of Neurophysiology* 92 (1): 236–54. <https://doi.org/10.1152/jn.01132.2003>.
- Das, Sudeshna, Madhumala K. Sadanandappa, Adrian Dervan, Aoife Larkin, John Anthony Lee, Indulekha P. Sudhakaran, Rashi Priya, et al. 2011. "Plasticity of Local GABAergic Interneurons Drives Olfactory Habituation." *Proceedings of the National Academy of Sciences* 108 (36): E646–54. <https://doi.org/10.1073/pnas.1106411108>.
- Fulton, Kara A., David Zimmerman, Aravi Samuel, Katrin Vogt, and Sandeep Robert Datta. 2024. "Common Principles for Odour Coding across Vertebrates and Invertebrates." *Nature Reviews Neuroscience* 25 (7): 453–72. <https://doi.org/10.1038/s41583-024-00822-0>.
- Grohmann, Lore, Wolfgang Blenau, Joachim Erber, Paul R. Ebert, Timo Strünker, and Arnd Baumann. 2003. "Molecular and Functional Characterization of an Octopamine Receptor from Honeybee (*Apis Mellifera*) Brain." *Journal of Neurochemistry* 86 (3): 725–35. <https://doi.org/10.1046/j.1471-4159.2003.01876.x>.
- Ito, Iori, Maxim Bazhenov, Rose Chik-ying Ong, Baranidharan Raman, and Mark Stopfer. 2009. "Frequency Transitions in Odor-Evoked Neural Oscillations." *Neuron* 64 (5): 692–706. <https://doi.org/10.1016/j.neuron.2009.10.004>.
- Lee, Hyun-Gwan, Chang-Soo Seong, Young-Cho Kim, Ronald L Davis, and Kyung-An Han. 2003. "Octopamine Receptor OAMB Is Required for Ovulation in *Drosophila*

- Melanogaster.” *Developmental Biology* 264 (1): 179–90.
<https://doi.org/10.1016/j.ydbio.2003.07.018>.
- Locatelli, Fernando F., Patricia C. Fernandez, and Brian H. Smith. 2016. “Learning about Natural Variation of Odor Mixtures Enhances Categorization in Early Olfactory Processing.” *The Journal of Experimental Biology* 219 (17): 2752–62.
<https://doi.org/10.1242/jeb.141465>.
- Locatelli, Fernando F., Patricia C. Fernandez, Francis Villareal, Kerem Muezzinoglu, Ramon Huerta, C. Giovanni Galizia, and Brian H. Smith. 2013. “Nonassociative Plasticity Alters Competitive Interactions among Mixture Components in Early Olfactory Processing.” *The European Journal of Neuroscience* 37 (1): 63–79.
<https://doi.org/10.1111/ejn.12021>.
- MaBouDi, HaDi, Hideaki Shimazaki, Martin Giurfa, and Lars Chittka. 2017. “Olfactory Learning without the Mushroom Bodies: Spiking Neural Network Models of the Honeybee Lateral Antennal Lobe Tract Reveal Its Capacities in Odour Memory Tasks of Varied Complexities.” Edited by Gabriel Kreiman. *PLOS Computational Biology* 13 (6): e1005551. <https://doi.org/10.1371/journal.pcbi.1005551>.
- Marachlian, Emiliano, Martin Klappenbach, and Fernando Locatelli. 2021. “Learning-Dependent Plasticity in the Antennal Lobe Improves Discrimination and Recognition of Odors in the Honeybee.” *Cell and Tissue Research* 383 (1): 165–75.
<https://doi.org/10.1007/s00441-020-03396-2>.
- Olsen, Shawn R., and Rachel I. Wilson. 2008. “Lateral Presynaptic Inhibition Mediates Gain Control in an Olfactory Circuit.” *Nature* 452 (7190): 956–60.
<https://doi.org/10.1038/nature06864>.
- Rein, Julia, Julie A. Mustard, Martin Strauch, Brian H. Smith, and C. Giovanni Galizia. 2013. “Octopamine Modulates Activity of Neural Networks in the Honey Bee Antennal Lobe.” *Journal of Comparative Physiology A* 199 (11): 947–62.
<https://doi.org/10.1007/s00359-013-0805-y>.
- Sinakevitch, Irina, Julie A. Mustard, and Brian H. Smith. 2011. “Distribution of the Octopamine Receptor AmOA1 in the Honey Bee Brain.” *PloS One* 6 (1): e14536.
<https://doi.org/10.1371/journal.pone.0014536>.